# HiPoNet: A Multi-View Simplicial Complex Network for High Dimensional Point-Cloud and Single-Cell data

Siddharth Viswanath[*1]    Hiren Madhu[*1]    Dhananjay Bhaskar[1]    Jake Kovalic[1]

David R. Johnson[2]    Christopher Tape[3]    Ian Adelstein[1]    Rex Ying[1]

Michael Perlmutter[2]    Smita Krishnaswamy[1]

[1] Yale University   [2] Boise State University   [3] University College London

[*]Equal Contribution    Correspondence: smita.krishnaswamy@yale.edu

## Abstract

In this paper, we propose HiPoNet, an end-to-end differentiable neural network for regression, classification, and representation learning on high-dimensional point clouds. Our work is motivated by single-cell data which can have very high-dimensionality – exceeding the capabilities of existing methods for point clouds which are mostly tailored for 3D data. Moreover, modern single-cell and spatial experiments now yield entire cohorts of datasets (i.e., one data set for every patient), necessitating models that can process large, high-dimensional point-clouds at scale. Most current approaches build a single nearest-neighbor graph, discarding important geometric and topological information. In contrast, HiPoNet models the point-cloud as a set of higher-order simplicial complexes, with each particular complex being created using a reweighting of features. This method thus generates multiple constructs corresponding to different views of high-dimensional data, which in biology offers the possibility of disentangling distinct cellular processes. It then employs simplicial wavelet transforms to extract multiscale features, capturing both local and global topology from each view. We show that geometric and topological information is preserved in this framework both theoretically and empirically. We showcase the utility of HiPoNet on point-cloud level tasks, involving classification and regression of entire point-clouds in data cohorts. Experimentally, we find that HiPoNet outperforms other point-cloud and graph-based models on single-cell data. We also apply HiPoNet to spatial transcriptomics datasets using spatial coordinates as one of the views. Overall, HiPoNet offers a robust and scalable solution for high-dimensional data analysis.

## 1 Introduction

High-dimensional point clouds, i.e., sets of points $\mathcal{X} = \{\mathbf{x}_i\}_{i=1}^n \subseteq \mathbb{R}^d$ now arise in many fields—most prominently single-cell analysis [55, 45, 12, 34, 1, 31], where using modern technologies such as mass cytometry or scRNA-seq, large cohorts of patients can now be measured producing several high-dimensional data. Further, technologies like perturb-seq enable the possibility of studying single-cell data under many conditions leading to comparable cohorts of datasets. Thus, machine learning techniques that once reasoned about data points and classified data points, now have to reason about collections of datasets. This serves as the motivation for HiPoNet illustrated in Figure 1.

Generally, methods like UMAP [36] or PHATE [38] reduce point clouds defined by single-cell data into an individual graph learned from all features (often in the tens of thousands). However, this single

graph may not specifically organize cells according to processes defined by subsets of dimensions. For instance, cells may be in a specific phase of the cell cycle which would be revealed by an organization based on cell cycle genes, or at a certain point in a differentiation process which would be revealed by stem cell genes. To address these limitations, HiPoNet utilizes multiple reweighted feature views. Additionally, existing neural network methods for point cloud data, such as PointNet [43] and its variants [44, 65, 58, 33] are primarily designed for 3D point clouds and rely on spatially localized features[1]. Hence, they are not well equipped to handle high-dimensional point clouds or disentangle processes from high-dimensional data. Therefore, there is a growing need to develop neural networks that can efficiently handle these diverse high-dimensional point clouds, ingest them seamlessly, encode the cellular processes, and perform various downstream machine learning tasks on them.

In HiPoNet, we use multiple views of the data, each of which is learned using a feature reweighting vector, thus potentially detangling and making implicit biological processes explicit. Furthermore, rather than modeling the data as a graph, we model it as a simplicial complex that captures higher-order relationships between cells, which HiPoNet analyzes using multiscale wavelets. Overall, we gain a rich representation that disentangles processes by capturing hierarchical relationships and separating overlapping biological processes, while preserving geometric and topological information of the underlying point clouds. Preserving this structure is important because it reflects the intrinsic geometry and topology of the data, providing a better interpretation of complex biological processes, and hence improving downstream analysis.

To capture the global structure of the point clouds, we use a wavelet-based multiscale message aggregation scheme rather than the message passing operations utilized in most common graph and simplicial neural networks. This choice is based on the observation that such networks struggle to capture long-range dependencies and multiscale relationships as well as work on the geometric scattering transform

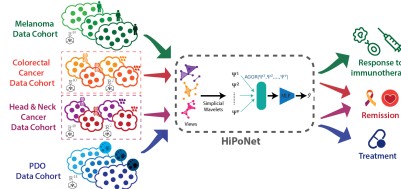

Figure 1: The HiPoNet pipeline.

[21, 41, 19, 20, 29, 59, 8, 62, 51] showing that wavelets may help remedy this issue. Although newer models like graph transformers [16] can capture long range dependencies, they often essentially ignore the graph geometry. When applied on these point clouds, they only work on a single graph and are do not preserve the intrinsic geometric structure of the point cloud. However, using diffusion wavelets, we are able to show theoretically that we capture the underlying geometry of the point cloud including the $0$-homology (connected components) and curvature. Empirically we show that we can predict persistence homology directly from our simplicial neural network, without explicitly computing topological signatures or reducing the data to those features. These properties that are preserved by HiPoNet are essential in its ability to perform classification tasks, such as response to immunotherapy or drug treatment response. The key features of HiPoNet include:

- **Learning multiple views the the data:** We introduce a learnable scaling mechanism to reweight each feature via a weighting vector, which we then use to learn a graph to organize the data by the processes represented by the particular view.

- **Higher-order constructions:** We learn simplicial complexes from the data views. Our neural networks utilize these complexes and retain geometric and topological information including volume, curvature, connected components, and holes in the data.

- **Multiscale Message Aggregation:** Rather than simple message passing schemes that smooth data, we use multiscale simplicial wavelets to aggregate features over the simplicial complex and construct simplicial scattering coefficients. This allows us to extract both local and global information from the data without oversmoothing.

We demonstrate the utility of HiPoNet on three diverse single-cell *data cohorts*, each comprising ensembles of datasets collected from multiple patients or conditions. The first involves melanoma [42] patients undergoing immunotherapy, where the task is to predict treatment response from multiplex ion beam imaging (MIBI) with 61K cells from 54 patients. The second features patient-derived organoids [45] (PDOs) from 12 colorectal cancer patients, grown under various treatments and co-culture conditions; comprising of 1.8M cells, and the task is to identify the applied treatment. The third cohort uses spatial transcriptomics data from around 500 cancer biopsies [61], capturing 40

---

[1]We discuss these techniques more in Appendix B .

protein markers per cell through CODEX technology along with spatial arrangements, comprising of about 558K cells across three data cohorts. These datasets highlight the scalability of HiPoNet for complex, high-dimensional biological point clouds, where complexity arises from the number of datasets rather than the number of points per dataset

## 2   Background

In this section, we present the mathematical foundations needed for our method. We model point clouds $\mathcal{X} = \{\mathbf{x}_1, \ldots, \mathbf{x}_n\}$ as simplicial complexes in order to capture higher-order geometric and topological structures. We will first introduce simplicial complexes and their algebraic representations, including boundary matrices and Laplacians. We then describe wavelet transforms and diffusion processes for extracting multi-scale topological features, and finally discuss random walks as efficient approximations of diffusion for scalable computation.

Given a finite set $\mathcal{X} = \{\mathbf{x}_1, \mathbf{x}_2, \ldots, \mathbf{x}_n\}$, a $k$-simplex, denoted as $\sigma_k = \{\mathbf{x}_{\sigma_k^0}, \mathbf{x}_{\sigma_k^1}, \ldots, \mathbf{x}_{\sigma_k^k}\}$, is a subset of $\mathcal{X}$ containing $k + 1$ points. If $\tau_{k-1}$ is a subset of $\sigma_k$ that contains $k$ points, we say that $\tau_{k-1}$ is a face of $\sigma_k$. A simplicial complex $\mathcal{S}$ is a finite collection of simplices that satisfies the principle of inclusion [5, 49]. That is, we have $\tau_{k-1} \in \mathcal{S}$ whenever if $\tau_{k-1}$ is a face of $\sigma_k$ for some $\sigma_k \in \mathcal{S}$. The order of a simplicial complex, $K$, is determined by the highest-order simplex contained within $\mathcal{S}$. Therefore a $K$-th order simplicial complex $\mathcal{S}$ contains simplices of orders $k = 0, 1, \ldots, K$. Observe that when $K = 1$, a simplicial complex is essentially equivalent to a graph, where the $0$-simplices and $1$-simplices correspond to the vertices and edges.

Features of simplices across all orders are denoted as $\mathbf{X} = \{\mathbf{X}_0, \mathbf{X}_1, \ldots, \mathbf{X}_K\}$, with $\mathbf{X}_k \in \mathbb{R}^{N_k \times D_k}$, where $N_k$ is the number of $k$ simplices and $D_k$ is the feature dimension of $k$-simplices. We note that we can consider both unoriented simplices or oriented simplices, where the orientation is defined via a fixed ordering on the vertices, i.e., $0$-complexes. (For details, please see [48] and the references within.) Except when otherwise specified, our theory and algorithms will apply to either case.

In a simplicial complex $\mathcal{S}$, a $k$-simplex $\sigma_k \in \mathcal{S}$ can have four types of neighbors:

- **Boundary adjacent neighbors or faces:** These are the $(k-1)$-simplices $\tau_{k-1}$ that are faces of $\sigma_k$, denoted as $\mathcal{N}_{\mathcal{B}}(\sigma_k) = \{\tau_{k-1} : \tau_{k-1} \subset \sigma_k\}$.

- **Coboundary adjacent neighbors or co-faces:** These are all $(k+1)$-simplices $\tau_{k+1}$ which have $\sigma_k$ as a face, denoted as $\mathcal{N}_{\mathcal{C}}(\sigma_k) = \{\tau_{k+1} : \sigma_k \subset \tau_{k+1}\}$.

- **Lower adjacent neighbors:** These are simplices $\tau_k$ of the same order as $\sigma_k$ that share a common face, $\rho_{k-1}$, represented as $\mathcal{N}_{\mathcal{L}}(\sigma_k) = \{\tau_k : \rho_{k-1} = \sigma_k \cap \tau_k, |\rho_{k-1}| = k - 1\}$.

- **Upper adjacent neighbors:** These are simplices, $\tau_k$ of the same order as $\sigma_k$ that are contained in a common $(k+1)$-simplex $\rho_{k+1} \in \mathcal{S}$, given by $\mathcal{N}_{\mathcal{U}}(\sigma_k) = \{\tau_k : \rho_{k+1} = \sigma_k \cup \tau_k, |\rho_{k+1}| = k + 1\}$.

We let $\mathcal{N}_1(\sigma_k) = \cup\{\sigma_k, \mathcal{N}_{\mathcal{B}}(\sigma_k), \mathcal{N}_{\mathcal{C}}(\sigma_k), \mathcal{N}_{\mathcal{L}}(\sigma_k), \mathcal{N}_{\mathcal{U}}(\sigma_k)\}$ denote the 1-hop neighborhood of a simplex $\sigma_k$, and for $m \geq 2$, we define the $m$-hop neighborhood of a simplex $\sigma_k$ is recursively as $\mathcal{N}_m(\sigma_k) = \cup_{\tau \in \mathcal{N}_{m-1}(\sigma_k)} \mathcal{N}_1(\tau)$ for $m \geq 2$.

The relationships between $k$-simplices and their faces are encoded in the boundary matrices, denoted as $\mathbf{B}_k \in \mathbb{R}^{N_{k-1} \times N_k}$, where $N_k$ is the number of simplices in $\mathcal{S}$ of order $k$. The $(i, j)$-th entry of $\mathbf{B}_k$ is non-zero if the $i$-th $(k-1)$-simplex is a face of the $j$-th $k$-simplex. For oriented simplices, these entries $\mathbf{B}_k$ take values $\pm 1$, reflecting their relative orientation. For unoriented simplices, all of the non-zero entries of $\mathbf{B}_k$ are equal to 1. We let $\mathbf{B} = \{\mathbf{B}_k\}_{k=1}^K$, denote the set of all boundary matrices.

We note that the boundary matrices are defined in terms of only boundary adjacent the and co-boundary adjacent neighbors. However, they may also be used to construct the Laplacians which encode information about lower and upper adjacent neighbors. Specifically, we define the lower and upper Laplcians by $\mathbf{L}_k^\ell = \mathbf{B}_k^\top \mathbf{B}_k$ and $\mathbf{L}_k^u = \mathbf{B}_{k+1} \mathbf{B}_{k+1}^\top$. We then define the $k$-Hodge Laplacian by:

$$\Delta_k = \mathbf{L}_k^\ell + \mathbf{L}_k^u, \tag{1}$$

(with $\mathbf{L}_0^\ell = \mathbf{L}_K^u = 0$). We note that for oriented simplices, $\Delta_0$ is merely the standard graph Laplacian.

## 2.1 Heat Diffusion and random walks on Simplicial Complexes

The heat equation on $k$-simplices [27] is given by:

$$\frac{\partial u_k(\sigma_k, t)}{\partial t} = -\Delta_k u_k(\sigma_k, t). \tag{2}$$

Analogous to classical notions of heat diffusion, it describes how the distribution of heat propagates over the simplicial complex over time. The use of $\Delta_k$ ensures that the diffusion process respects the higher-order connectivity of the complex, encoding its topological structure.

The intuition behind much of our methods is to use the heat equation to uncover the topological structure of our point clouds and simplicial complexes, analogous to manifold learning algorithms such as Diffusion Maps [14]. However, in our actual algorithm, we will use random walk matrices, which we interpret as computationally efficient proxies for heat diffusion (similar to [14] which uses a diffusion matrix as a discrete approximation of the heat kernel on the underlying data manifold). Analogous to heat equation, random walks allow information to spread, i.e., diffuse, over the simplicial complex. However, they allow for efficient implementation via sparse matrix-vector multiplications.

We note that simplicial random walks extend the concept of traditional random walks from graphs to simplicial complexes, enabling a richer representation of the data which is able to encode higher-order features. Unlike standard graph-based random walks, which model transitions between vertices, simplicial random walks capture transitions between $k$-simplices, thereby incorporating higher-order interactions. Formally, the simplicial random walk is represented by the transition matrix $\mathbf{P}_k$, which describes the probability of transitioning between $k$-simplices. For unoriented simplices, we define:

$$\mathbf{P}_k = \Delta_k \mathbf{D}_k^{-1}, \tag{3}$$

where $\mathbf{D}_k = \mathrm{diag}(\Delta_k \mathbf{1})$ is a diagonal matrix that normalizes the transition probabilities, ensuring that the columns of $\mathbf{P}_k$ sum to one. In the case of oriented simplices, defining $\mathbf{P}_k$ is non-trivial since the entries of $\Delta_k$ may be either positive or negative. Here, for the sake of simplicity, we will define $\mathbf{P}_k$ the same way as in the unoriented case, but with first taking the entrywise absolute values of $\Delta_k$, i.e., $\mathbf{P}_k = |\Delta_k| \mathbf{D}_k^{-1}$, $\mathbf{D}_k = \mathrm{diag}(|\Delta_k| \mathbf{1})$. Alternatively, in the case $k = 0$ or 1, one could define $\mathbf{P}_k$ in terms of a suitably normalized version of the Hodge Laplacian. (For details, please see [50].)

## 3 Methodology

We now describe the proposed method, HiPoNet, a High-dimensional POint cloud neural NETwork designed to learn expressive representations of an input point set $\mathcal{X} = \{\mathbf{x}_1, \mathbf{x}_2, \ldots, \mathbf{x}_n\} \in \mathbb{R}^d$. As summarized in Algorithm 1, our framework comprises three primary steps:

1. **Multi-view creation via learnable feature weighting:** We learn multiple sets of weights $\alpha^{(v)}, 1 \leq v \leq V$, to highlight different feature dimensions to create reweighted point clouds $\tilde{\mathcal{X}}^{(v)}$, which are effectively different views of the data.

2. **Simplicial complex-based point-cloud modeling:** From each $\tilde{\mathcal{X}}^{(v)}$, we construct a Vietoris-Rips complex $\mathcal{S}^{(v)}$ [57]. This approach allows us to capture not only pairwise relationships but also higher-order interactions within the data.

3. **Multi-scale Feature Extraction via Wavelets:** For each simplicial complex $\mathcal{S}^{(v)}$, we use wavelets to construct simplicial scattering coefficients which form a multi-scale embeddings $\Psi^{(v)}$, that capture both local and global relationships. These embeddings are then concatenated into a single representation $\Phi$ and are passed to a multilayer perceptron to yield the final predictions $\hat{\mathbf{y}}$.

By combining learnable multi-view feature weighting, simplicial complex construction, and simplicial wavelet transforms, HiPoNet leverages multiple perspectives of the underlying point cloud, at multiple scales to deliver robust representations for downstream classification and prediction.

**Multi-view creation via learnable feature weighting:** In many scenarios, such as scRNA-seq, feature vectors can encode critical properties, such as gene expression levels. However, not all dimensions (genes) contribute equally to downstream tasks, and irrelevant or noisy features can

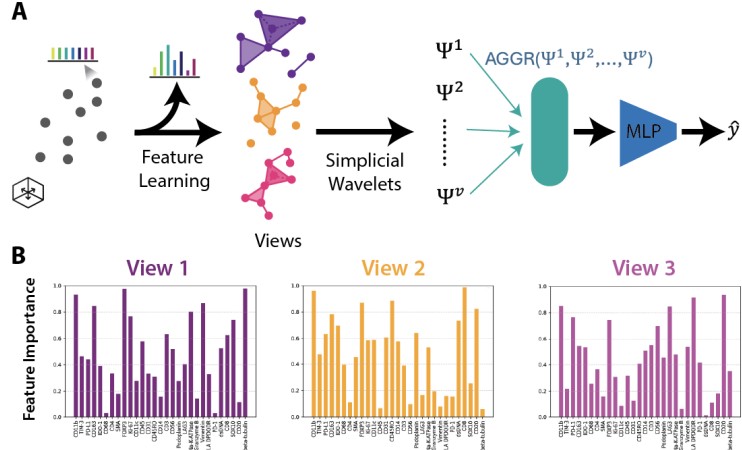

Figure 2: (A) The HiPoNet architecture. (B) Feature weight visualized across three learned views

---

**Algorithm 1:** HiPoNet

**Input** : Point cloud $\mathcal{X}$, kernel bandwidth $\sigma$, number of views $V$.
        Learnable Parameters: learnable weights $\alpha^{(v)}$, MLP $\mathbf{z}$.

**Output** : Predictions $\hat{\mathbf{y}}$.

1 **for** $v = 1$ **to** $V$ **do**

2     $\tilde{\mathbf{x}}_i^{(v)} = \alpha^{(v)} \odot \mathbf{x}_i$ for $1 \leq i \leq n$ ;

3     $\tilde{\mathcal{X}}^{(v)} = \{\tilde{\mathbf{x}}_i^{(v)}\}_{i=1}^n$ ;

4     $\mathcal{S}^{(v)} = \text{Vietoris-Rips}(\tilde{\mathcal{X}}^{(v)}, \epsilon)$ ;

5     Compute boundary matrices $\mathbf{B}^{(v)}$ from simplices in $\mathcal{S}^{(v)}$ ;

6     Set $\mathbf{X}_0^{(v)} = \tilde{\mathcal{X}}^{(v)}$                 // Vertex features ;

7     **for** $k = 0$ **to** $K - 1$ **do**

8        $\mathbf{X}_{k+1}^{(v)} = (\mathbf{B}_k^{(v)})^\top \mathbf{X}_k^{(v)}$       // Features for higher-order simplices

9     **for** $k = 0$ **to** $K$ **do**

10        Compute transition matrix: $\mathbf{P}_k^{(v)} = \Delta_k^{(v)} \mathbf{D}_k^{-1}$ ;

11        **for** $j = 1$ **to** $J$ **do**

12           $\Psi_k^{(v,j)} \mathbf{X}_k^{(v)} = \left( \left(\mathbf{P}_k^{(v)}\right)^{2^j} - \left(\mathbf{P}_k^{(v)}\right)^{2^{j-1}} \right) \mathbf{X}_k^{(v)}$ ;

13        Compute Scattering Coefficients:

14        $S_k^v[j] \mathbf{X}_k^{(v)} = |\Psi_k^{(v,j)} \mathbf{X}_k^{(v)}|$ and $S_k^v[j, j'] \mathbf{X}_k^{(v)} = ||\Psi_k^{(v,j')}| \Psi_k^{(v,j)} \mathbf{X}_k^{(v)}||$

15     Aggregate features across simplex orders: $|S(\mathbf{B}^{(v)}, \mathbf{X}^{(v)})| = \left\{ |S_0^{(v)} \mathbf{X}_0^{(v)}|, \ldots, |S_K^{(v)} \mathbf{X}_K^{(v)}| \right\}$ ;

16 **Final Aggregation and Prediction:**

17 $\Phi = \text{Aggr}\{S(\mathbf{B}^{(1)}, \mathbf{X}^{(1)}), \ldots, S(\mathbf{B}^{(V)}, \mathbf{X}^{(V)})\}$ ;

18 $\hat{\mathbf{y}} = \mathbf{z}(\Phi)$ ;

19 **return** $\hat{\mathbf{y}}$

---

impair model performance, especially in high-dimensional settings. To address this, we introduce a learnable weighting mechanism that dynamically adjusts the weight of each feature dimension. Concretely, we define $V$ distinct weight vectors $\alpha^{(v)}$, $1 \leq v \leq V$, each as a parameter of the model. For the $v$-th view, each point $\mathbf{x}_i$ is rescaled to

$$\tilde{\mathbf{x}}_i^{(v)} = \alpha^{(v)} \odot \mathbf{x}_i = [\alpha_1^{(v)} x_{i1},\ \alpha_2^{(v)} x_{i2}, \ldots, \alpha_d^{(v)} x_{id}],$$

where $\odot$ denotes the Hadamard (elementwise) product. We let $\tilde{\mathcal{X}}^{(v)} = \{\tilde{\mathbf{x}}_1^{(v)}, \tilde{\mathbf{x}}_2^{(v)}, \ldots, \tilde{\mathbf{x}}_n^{(v)}\}$ denote the reweighted point cloud. The reweighting operations amplify the more informative dimensions for the task at hand while downplaying less relevant features, effectively reducing the need for manual feature engineering. In the context of single-cell data, where the feature space can be vast,

such automated reweighting may be crucial for isolating key gene expressions that drive improved downstream performance. The learned weights can also be used for other biological insights, such as which gene markers are important for cancer diagnosis as illustrated in Figure 2B.

**Simplicial learning from piont cloud data** For each reweighted set $\tilde{\mathcal{X}}^{(v)}$, we create a simplicial complex $\mathcal{S}^{(v)}$. To do this, we first define kernelized distances between two points $\tilde{\mathbf{x}}_i^{(v)}$ and $\tilde{\mathbf{x}}_j^{(v)}$ as:

$$d_{i,j}^{(v)} = \exp\left(\frac{-\|\tilde{\mathbf{x}}_i^{(v)} - \tilde{\mathbf{x}}_j^{(v)}\|_2^2}{2\sigma^2}\right).$$

We then fix a scale parameter, $\epsilon$, and construct the Vietoris-Rips Complex $\mathcal{S}^{(v)}$, a simplicial complex defined by the rule that $\sigma_k = \{\mathbf{X}_{i_0}^{(v)}, \mathbf{X}_{i_1}^{(v)}, \ldots, \mathbf{X}_{i_k}^{(v)}\}$ is an element of $\mathcal{S}^{(v)}$ if $d_{i_p,i_q}^{(v)} \leq \epsilon$ for all $1 \leq p, q \leq k$. After constructing the simpicial complex $\mathcal{S}^{(v)}$, we construct the boundary operators $\mathbf{B}^{(v)}$. We then define feature matrices by $\mathbf{X}_0^{(v)} = \tilde{\mathcal{X}}^{(v)}$ (i.e., the $i$-th row of $\mathbf{X}_0^{(v)}$ is $\tilde{\mathbf{x}}_i^{(v)}$) and then $\mathbf{X}_{k+1}^{(v)} = (\mathbf{B}_k^{(v)})^T \mathbf{X}_k^{(v)}$. In our code, we use a custom implementation of the Vietoris-Rips filtration that enables us to make simplicial construction differentiable, making HiPoNet end-to-end trainable.

We repeat this process $V$ times, and construct $\mathbf{B}^{(v)}$ for each view $1 \leq v \leq V$. By constructing an ensemble of simplicial complexes $\mathcal{S}^{(1)}, \mathcal{S}^{(2)}, \ldots, \mathcal{S}^{(V)}$, our method integrates $V$ distinct perspectives of the original high-dimensional point cloud. Constructing a simplicial complex $\mathcal{S}^{(v)}$ for each reweighted set $\tilde{\mathcal{X}}^{(v)}$ allows the model to disentangle and analyze distinct subspaces or biological processes captured by each view. For instance, each complex may focus on a different cellular process, capturing unique local and global structures that is overlooked when relying on a single representation. Consequently, subsequent learning stages benefit from a richer set of structural cues, enhancing the overall representation quality for downstream tasks.

**Multi-scale feature extraction via simplicial wavelets** After the simplicial complexes have been constructed, we can leverage geometric deep learning methods to compute representations for the point cloud. Although message passing neural networks (MPNNs) for simplicial complexes [6, 9, 23] have been introduced, they often suffer from oversmoothing effects [4], where representations become indistinguishable after multiple layers. In addition, oversquashing [2] is another constraint of such message-passing frameworks. As the receptive field of each node grows, large amounts of information from distant nodes are compressed or "squashed" into a fixed-size vector, leading to loss of important information between the nodes. This limits the ability of MPNNs to model long-range dependencies.

In order to address these shortcomings, we employ the simplicial wavelet transform [35, 46, 47] (SWT). The SWT is a multi-scale feature extractor for processing a signal $\mathbf{X}$ defined on the simplices. These wavelets are based on diffusion wavelets introduced in [13] which utilize random walk matrices at different time scales. In SWT, the random walk matrices are calculated first as per Eq 3. Then, features are iteratively aggregated over all the neighborhoods and stored for each scale $j$. Finally, a difference between different scales is taken to extract multiscale features.

The random walk diffusion matrix plays a crucial role in the SWT. For each view $v$, we interpret each $\left(\mathbf{P}_k^{(v)}\right)^j \mathbf{X}_k^{(v)}$ as diffused node features at scale $j$. Our wavelets, defined below, can be thought of as measuring the differences between these diffused features at different scales. Alternatively, from a signal processing perspective, these wavelets can be interpretted as a band-pass filters, with each $j$ highlighting different frequency bands within the signal.

Formally, the simplicial wavelet transform at scale $j$, $0 \leq j \leq J$, is defined as the difference between diffusion at consecutive scales:

$$\Psi_k^{(v,j)} \mathbf{X}_k^{(v)} = \left(\left(\mathbf{P}_k^{(v)}\right)^j - \left(\mathbf{P}_k^{(v)}\right)^{j-1}\right) \mathbf{X}_k^{(v)}. \tag{4}$$

After applying SWT for multiple scales $j$, we apply a non-linearity, which we choose to be the absolute value $|\cdot|$, inspired by the geometric scattering transform [22, 19, 66, 41], a multilayered feedforward network constructed using diffusion wavelets on graphs. We then repeat this process with second wavelet, at a different scale $j' > j$, and then another absolute value. This yields first- and second-order *simplicial scattering coefficients* of the form $S_k^v[j]\mathbf{X}_k^{(v)} = |\Psi_k^{(v,j)}\mathbf{X}_k^{(v)}|$ and $S_k^v[j,j']\mathbf{X}_k^{(v)} =$

$|\Psi_k^{(v,j')}|\Psi_k^{(v,j)}\mathbf{X}_k^{(v)}||$. We then denote the collection of all first- and second-order scattering coefficients for each view by $|S(\mathbf{B}^{(v)}, \mathbf{X}^{(v)})| = \{S_k^v[j]\mathbf{X}_k^{(v)}\}_{\substack{0 \le j \le J, \\ 1 \le k \le K}} \cup \{S_k^v[j,j']\mathbf{X}_k^{(v)}\}_{\substack{0 \le j \le j' \le J, \\ 1 \le k \le K}}$. After extracting features for each view $v$, we aggregate the features over views as:

$$\Phi = \mathrm{Aggr}\{S(\mathbf{B}^{(1)}, \mathbf{X}^{(1)}), S(\mathbf{B}^{(2)}, \mathbf{X}^{(2)}), \dots, S(\mathbf{B}^{(V)}, \mathbf{X}^{(V)})\}.$$

$\Phi$ is then processed by a multilayer perceptron $\mathbf{z}$ to generate our final predictions, $\hat{\mathbf{y}} = \mathbf{z}(\Phi)$.

## 4   Theoretical results

In this section, we provide theoretical motivation for our model. Specifically, we first show that heat diffusion on simplices captures the 0-homology of the point cloud. Then, we show that simplicial complexes can be expanded into a graph, and the heat equation on the simplicial complex agrees with the heat equation on the equivalent graph. Then, we show that the diffusion operators can capture geodesic distances. Proofs, as well as more detailed theorem statements, are provided in the appendix.

**Heat Diffusion and Connectivity.** A simplicial complex is said to be connected if any two simplices can be linked by a sequence of simplices that share common faces. More formally, if for any two simplices $\sigma$ and $\tau$, there exists $m \ge 0$ such that $\tau \in \mathcal{N}_m(\sigma)$. If a simplicial complex is not connected, it can be decomposed into a union of disjoint connected components, each of which is a maximal connected subcomplex.

The following theorem demonstrates that the solution to the heat equation on a simplicial complex remains confined to the connected components where the initial condition is supported. This reflects the intuitive idea that heat cannot diffuse across disconnected regions of the complex. It also aligns with the concept of 0-homology in algebraic topology since the connected components of a simplicial complex correspond to the generators of its 0-homology group. Thus, the heat equation's confinement to connected components ensures that diffusion dynamics respect the 0-homology structure of the simplicial complex. The proof of Theorem 4.1 is available in Appendix C.

**Theorem 4.1.** *The heat equations, Eq 2, respect the 0-homology structure of the simplicial complex.*

*Proof Sketch.* The heat equation is governed by $\Delta_k$, which acts locally by coupling neighbors. This ensures that heat diffusion cannot propagate between connected components. $\qquad\square$

**Heat Diffusion on Simplicial Graphs.** In this section, we demonstrate that the solution to the heat equation on a simplicial complex $\mathcal{S}$ agrees with the solution to the heat equation on an associated simplicial graph, i.e., a graph where the vertices represent simplices of all orders and edges encode upper and lower adjacencies. This construction allows us to equate heat diffusion on the simplicial complex as heat diffusion on an associated graph, providing a simplified framework for analysis. The proof of Theorem 4.3 is in Appendix D.

**Definition 4.2** (Simplicial Graph). Let $\mathcal{S}$ be a simplicial complex. The associated simplicial graph $\mathcal{G}(\mathcal{S})$ is a graph whose vertices are given by $V(\mathcal{G}) = \mathcal{S}$ and whose edge set $E(\mathcal{G})$ consists of pairs of simplices which are either upper or lower adjacent, as defined in Section 2, that is,

$$E(\mathcal{G}) = \{\{\sigma, \sigma'\} \mid \sigma, \sigma' \in V(\mathcal{G}), \sigma' \in \mathcal{N}_{\mathcal{L}}(\sigma) \cup \mathcal{N}_{\mathcal{U}}(\sigma)\}.$$

**Theorem 4.3.** *The heat equation on a simplicial complex $\mathcal{S}$ agrees with the heat equation on the associated simplicial graph $\mathcal{G}(\mathcal{S})$, defined in terms of $\Delta_{\mathcal{G}} = B(\mathcal{G}(\mathcal{S}))B(\mathcal{G}(\mathcal{S}))^T$, where $B(\mathcal{G}(\mathcal{S}))$ is the incidence matrix of $\mathcal{G}(\mathcal{S})$.*

*Proof Sketch.* The graph Laplacian $\Delta_{\mathcal{G}}$ on the simplicial graph can be represented as a block diagonal matrix with the Hodge Laplacians $\Delta_k$ as the diagonal blocks. $\qquad\square$

**Heat Solution Captures Geometry.** Most of our experiments focus on single-cell data which are known to satisfy the manifold hypotheses [28, 24, 17, 55, 1]. That is, the data points $\mathbf{x}_i$ lie upon some low-dimensional manifold. More specifically, we interpret each of the transformed point clouds $\tilde{\mathcal{X}}^{(v)}$ as corresponding to different submanifolds of some global cellular manifold. The following theorem establishes that the diffusion operator on the simplicial complex can approximate geodesic distances on these underlying manifolds. The proof of Theorem 4.4 is in Appendix E.

**Theorem 4.4.** *Assume that a transformed point cloud $\tilde{\mathcal{X}}^{(v)}$ lies upon a Riemannian manifold $\mathcal{M}$. Then, the $0$-th order Hodge Laplacian $\Delta_0$ on the associated oriented simplicial complex can approximate geodesic distances on the underlying manifold.*

*Proof Sketch.* The result follows from Varadhan's formula [53], which shows that distances may be computed via the heat kernel, as well as Theorem 3 of [15], which analyzes the convergence of the graph heat kernel to the manifold heat kernel as well as Theorem 4.3 which relates the graph heat equation to the simplicial complex heat equation. □

This theorem formalizes the link between diffusion processes, e.g., the heat equation, and the geometric structure of the data manifold. Notably, computation of geodesic distances on manifolds is computationally expensive and often infeasible in high dimensions. By using diffusion-based approximations, we can efficiently estimate geodesic distances with discrete operators. Additionally, we note that there is a long literature (dating back to at least [13]) relating the Laplacian and the diffusion operator to the geometry of the data manifold, including quantities such as curvature [7]. Most relevant to our work are the results which relate the asymptotic expansion of the trace of the heat kernel to geometric quantities like dimension, volume, and total scalar curvature (see for instance [37]). Indeed, following immediately from Theorems 4.3 and 4.4 we have:

**Corollary 4.5.** *The equivalence between the heat kernel on $\mathcal{S}$ and $\mathcal{G}(\mathcal{S})$ implies the equivalence of dimension, volume, and total scalar curvature on the respective underlying data manifolds.*

The proof of Corollary 4.5 is available in Appendix Section F. We use Weyl's law [60] and the eigenvalue comparison theorem [11] to prove the equivalence.

## 5 Empirical Results

We compare the performance of HiPoNet with KNN-GNNs (i.e., GNNs on KNN graphs) as well as PointNet++ and it's variants are described in Appendix H.1. The KNN-based graph neural networks (GCN [30], SAGE [25], GAT [54], GIN [63] and Graph Transformer [16], TopoGNN [26]) are presented in the first block, followed by point cloud and topology oriented models (DGCNN [58], PointNet++ [44], PointTransformer [65], TopoAE [39], GCNN [40], HGCNN [18]), and finally the proposed HiPoNet. We present the mean and standard deviation over 5 seeds[2]. A bolded score indicates the highest overall performance, whereas an underlined value identifies the second-best performance. Unless otherwise stated, we use unoriented simplices in our experiments. Details on the data cohorts considered in our experiments are provided in Appendix G. Hyperparameters are described in Appendix H. We provide a discussion of computational complexity in Appendix I. Discussion of limitations and societal impact is provided in Appendix Sections L and K, respectively.

**Topology and Geometry Prediction.** We evaluate HiPoNet on the task of predicting persistence features of the point clouds, which provides a topological summary of the dataset. The ground truth for persistence features was calculated by calculating using persistence diagrams. We note that the purpose of these experiments is not to develop a new method of computing persistence features. Instead, it is to show that HiPoNet has the expressivity needed in order to compute such features, indicating that it is a viable method for tasks which require a network to capture the underlying topology of the data.

The results in Table 1 indicate that HiPoNet outperforms baseline methods, achieving the lowest MSE across both datasets. Notably, KNN-based models perform competitively, with KNN-SAGE and KNN-GIN showing relatively lower errors compared to other GNN-based approaches. However, point-cloud-based models (DGCNN, PointNet++, and PointTransformer) exhibit significantly higher error values, suggesting that they struggle to capture the necessary

| Model | Melanoma | PDO |
|---|---|---|
| KNN-GCN | $1.0683 \pm 0.002$ | $1.0454 \pm 0.018$ |
| KNN-SAGE | $0.734 \pm 0.031$ | $1.0338 \pm 0.010$ |
| KNN-GAT | $1.101 \pm 0.028$ | $\underline{1.068 \pm 0.008}$ |
| KNN-GIN | $0.850 \pm 0.061$ | $1.2605 \pm 0.006$ |
| KNN-GraphTransformer | $1.274 \pm 0.038$ | $1.3754 \pm 0.006$ |
| DGCNN | $28.412 \pm 0.001$ | $1353.0262 \pm 1.12$ |
| PointNet++ | $28.418 \pm 0.001$ | $28.417 \pm 0.005$ |
| PointTransformer | $28.422 \pm 0.014$ | $28.41 \pm 0.003$ |
| HiPoNet | $\mathbf{0.633 \pm 0.043}$ | $\mathbf{0.4046 \pm 0.006}$ |

Table 1: MSE on prediction of persistence features.

---

[2]The code is available at `https://github.com/KrishnaswamyLab/HiPoNet`.

| Data cohort | Nuumber of cells | Total # Datasets | Task | Data Modality |
|---|---|---|---|---|
| Melanoma patient samples | 61K | 54 | Response to Immunotherapy | MIBI |
| Patient-derived organoids (PDOs) | 1.8M | 1625 | Treatment Administered | Mass Cytometry |
| Charville | 196K | 196 | Outcome to chemotherapy | |
| | 196K | 196 | Cancer recurrence | CODEX |
| UPMC | 308K | 308 | Outcome to chemotherapy | |
| | 308K | 308 | Cancer recurrence | |
| DFCI | 54K | 54 | Outcome to chemotherapy | |

Table 2: Information about single-cell data

topological features for persistence feature prediction on these data sets, perhaps because they were designed with 3D point clouds in mind, rather than high-dimensional point clouds.

**Single-cell data classification.** Next we assess the performance of HiPoNet on classifiying point clouds from single-cell data cohorts, described in Table 2. (Detailed descriptions of this data is provided in the Appendix G.) As mentioned in the introduction, we emphasize that each data cohort is a measurement of thousands cells on a large cohort of patients and the label is prediction of outcome (of disease or treatment). As we can see in Table 4, HiPoNet outperforms all the graph-based, topological, and point cloud methods in Melanoma. On the PDO data, it is second to TopoGNN. However, HiPoNet is much more consistent than TopoGNN having a standard deviation of $0.94\%$ in comparison to TopoGNNs standard deviation of $16.15\%$.

**Application to spatial transcriptomics data.** In addition to the experiments on single-cell data, we further demonstrate an application of HiPoNet in analyzing spatial transcriptomics data. Unlike previous experiments where multiple views were constructed in the same manner, here we construct two views by different methods: one capturing spatial proximity of cells and the other encoding gene expression similarity. Table 3 compares the predictive performance of various models on spatial transcriptomics data drawn from four different cohorts: DFCI, Charville, UPMC, and Melanoma. DFCI and Melanoma include one task-outcome prediction-while Charville and UPMC includes two tasks—either outcome prediction or recurrence prediction. Notably, HiPoNet achieves the top results in most settings, outperforming KNN-based graph neural networks (GCN, SAGE, GAT, GIN), PointNet++, and the KNN-GraphTransformer.

| Data | DFCI | Charville | | UPMC | | Melanoma |
|---|---|---|---|---|---|---|
| Task | Outcome | Outcome | Recurrence | Outcome | Recurrence | Response |
| KNN-GCN | $0.597 \pm 0.049$ | $0.547 \pm 0.010$ | $0.642 \pm 0.056$ | $\mathbf{0.668 \pm 0.032}$ | $0.5 \pm 0.0$ | $0.606 \pm 0.13$ |
| KNN-SAGE | $\underline{0.82 \pm 0.040}$ | $0.618 \pm 0.021$ | $0.581 \pm 0.016$ | $0.631 \pm 0.013$ | $0.5 \pm 0.0$ | $0.572 \pm 0.05$ |
| KNN-GAT | $0.557 \pm 0.047$ | $\underline{0.675 \pm 0.051}$ | $0.530 \pm 0.032$ | $0.647 \pm 0.029$ | $0.5 \pm 0.0$ | $0.567 \pm 0.05$ |
| KNN-GIN | $0.700 \pm 0.048$ | $0.609 \pm 0.020$ | $0.624 \pm 0.045$ | $0.663 \pm 0.015$ | $\underline{0.514 \pm 0.02}$ | $0.567 \pm 0.06$ |
| KNN-GraphTransformer | $0.668 \pm 0.051$ | $0.578 \pm 0.040$ | $0.5 \pm 0.0$ | $0.629 \pm 0.00$ | $0.5 \pm 0.0$ | $0.528 \pm 0.04$ |
| PointNet++ | $0.491 \pm 0.057$ | $0.451 \pm 0.078$ | $0.499 \pm 0.001$ | $0.506 \pm 0.01$ | $0.495 \pm 0.00$ | $0.503 \pm 0.00$ |
| HiPoNet | $\mathbf{0.916 \pm 0.03}$ | $\mathbf{0.681 \pm 0.012}$ | $\mathbf{0.681 \pm 0.01}$ | $\underline{0.665 \pm 0.01}$ | $\mathbf{0.6044 \pm 0.0}$ | $\mathbf{0.732 \pm 0.01}$ |

Table 3: AUC ROC on Spatial Transcriptomics Classification

**Intepretability of Learned Features.** HiPoNet offers high degree of interpretability enabled by its multi-view and learnable feature weighting architecture that learns meaningful representations from high-dimensional point cloud data. As seen in the three bar plots in Figure 2(B), each plot corresponds to a different learned view or representation of the 30 protein expression markers in the melanoma single-cell classification task, with the y-axis including the learning importance of each marker in that specific view. Biologically meaningful markers like CD11b,

| Model | Melanoma | PDO |
|---|---|---|
| KNN-GCN | $72.72 \pm 5.45$ | $53.66 \pm 0.70$ |
| KNN-SAGE | $76.36 \pm 6.03$ | $55.89 \pm 0.83$ |
| KNN-GAT | $61.81 \pm 4.45$ | $52.56 \pm 10.06$ |
| KNN-GIN | $\underline{85.45 \pm 3.63}$ | $57.02 \pm 1.20$ |
| KNN-GraphTransformer | $80.00 \pm 8.90$ | $39.46 \pm 2.78$ |
| TopoGNN | $\underline{88.18 \pm 8.19}$ | $\mathbf{79.90 \pm 16.15}$ |
| HGNN | $80.00 \pm 7.6$ | $38.76 \pm 0.87$ |
| DGCNN | $63.33 \pm 12.47$ | $40.00 \pm 4.36$ |
| PointNet++ | $45.00 \pm 22.42$ | $45.24 \pm 2.08$ |
| PointTransformer | $79.99 \pm 6.24$ | $30.00 \pm 4.70$ |
| TopoAE | $52.73 \pm 11.35$ | $15.71 \pm 0.87$ |
| GCNN | $63.63 \pm 0.0$ | $29.34 \pm 1.03$ |
| HiPoNet | $\mathbf{90.90 \pm 4.92}$ | $\underline{77.38 \pm 0.94}$ |

Table 4: Accuracies on classification tasks.

CD118, and FOXP3 have consistently high importance across the different views, aligning with their roles in immune regulation within the tumor microenvironment and promoting immune evasion in tumors.

**Mitigation of Oversmoothing.** To empirically examine whether our modeling choice mitigates oversmoothing, we conducted a comparative analysis of Dirichlet energy across various methods. As established by [10], Dirichlet energy provides a principled measure of the expressiveness of node representations, where lower values indicate oversmoothed embeddings, and higher values reflect

greater feature variation across graph neighborhoods. We computed the Dirichlet energy of node representations produced by different models.

As seen in Table 5, results indicate that our model, HiPoNet, exhibits substantially higher Dirichlet energy than traditional message-passing networks. This supports the claim that HiPoNet mitigates oversmoothing and retains richer, more expressive node-level information. Notably, the GCN and diffusion-based models

| Model | Dirichlet Energy |
|-------|-----------------|
| GCN | $1.669 \pm 0.13$ |
| GAT | $2.694 \pm 0.49$ |
| Graph Transformer | $3.811 \pm 0.80$ |
| Diffusion-based GNN | $0.584 \pm 0.00$ |
| GWT | $15.807 \pm 0.00$ |
| **HiPoNet (ours)** | **$21.033 \pm 6.25$** |

Table 5: Dirichlet Energy across different models.

exhibit particularly low energy, consistent with oversmoothed representations. Thus, this empirical evidence reinforces the motivation for using wavelet transforms, which allow HiPoNet to capture multi-scale variation and higher-frequency components in the signal that are typically suppressed in standard GNN architectures.

**Ablations.** We provide ablations in Appendix J. Table A4 shows that increasing the number of views improves performance, with four views yielding the best results. In Table A5, we observe that removing multi-view learning or simplicial modeling leads to the largest performance drops, confirming their critical importance. Table A6 presents a sensitivity analysis of the Vietoris–Rips threshold, showing that the optimal threshold varies by dataset and significantly affects accuracy. In Table A7, we analyze the effect of kernel bandwidth, finding that careful tuning is required to avoid oversmoothing. Finally, Table A8 demonstrates that while 1st order simplices (essentially graphs) are sufficient for some data sets, incorporating higher-order simplices improves performance for others, particularly in predicting clinical outcomes.

## 6 Conclusion

In this work, we introduced HiPoNet, a novel neural network architecture designed for high-dimensional point cloud datasets. By leveraging multiple views, higher-order constructs, and multi-scale wavelet transforms, HiPoNet effectively captures complex geometric and topological structures inherent in biological data. Our experiments demonstrate that HiPoNet learns distinct, meaningful representations tailored to each dataset and significantly improves downstream analysis across diverse single-cell and spatial transcriptomics cohorts. These results highlight the importance of integrating higher-order and multi-scale information to overcome the limitations of existing point cloud and graph-based models.

## 7 Acknowledgements

D.B. received funding from the Yale - Boehringer Ingelheim Biomedical Data Science Fellowship and the Kavli Institute for Neuroscience Postdoctoral Fellowship. M.P. acknowledges funding from The National Science Foundation under grant number OIA-2242769. S.K. is funded in part by the NIH (NIGMSR01GM135929, R01GM130847), NSF CAREER award IIS-2047856, NSF IIS-2403317, NSF DMS-2327211 and NSF CISE-2403317. S.K is also funded by the Sloan Fellowship FG-2021-15883, the Novo Nordisk grant GR112933. S.K. and M.P. also acknowledge funding from NSF DMS-2327211.

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

## A Notations

Table A1: Notations used throughout the paper.

| Notation | Description |
|---|---|
| $\mathcal{X} = \{\mathbf{x}_1, \ldots, \mathbf{x}_n\}$ | Original point cloud with $n$ data points |
| $d$ | Input feature dimension (e.g., number of genes or markers) |
| $V$ | Number of distinct views |
| $\alpha^{(v)} \in \mathbb{R}^d$ | Learnable weight vector for view $v$ |
| $\tilde{\mathbf{x}}_i^{(v)}$ | Reweighted feature vector for point $i$ in view $v$ |
| $\tilde{\mathcal{X}}^{(v)}$ | Reweighted point cloud for view $v$ |
| $d_{ij}^{(v)}$ | Gaussian Kernel distance between points $i$ and $j$ in view $v$ |
| $\epsilon$ | Scale parameter for Vietoris-Rips complex construction |
| $\mathcal{S}^{(v)}$ | Simplicial complex for view $v$ |
| $K$ | Maximum simplex order in the complex |
| $\sigma_k$ | A $k$-simplex in the complex |
| $N_k$ | Number of $k$-simplices |
| $\mathbf{X}_k^{(v)} \in \mathbb{R}^{N_k \times D_k}$ | Feature matrix for $k$-simplices in view $v$ |
| $\mathbf{B}_k^{(v)}$ | Boundary matrix for order-$k$ simplices in view $v$ |
| $\Delta_k^{(v)}$ | Hodge Laplacian for $k$-simplices in view $v$ |
| $\mathbf{P}_k^{(v)}$ | Simplicial random walk matrix for $k$-simplices in view $v$ |
| $J$ | Number of diffusion scales used in the scattering transform |
| $\Psi_k^{(v,j)}$ | Simplicial wavelet at scale $j$ for order-$k$ simplices in view $v$ |
| $S_k^v[j]\mathbf{X}_k^{(v)}$ | First-order scattering coefficients |
| $S_k^v[j,j']\mathbf{X}_k^{(v)}$ | Second-order scattering coefficients |
| $S(\mathbf{B}^{(v)}, \mathbf{X}^{(v)})$ | Collection of all scattering features for view $v$ |
| $\Phi$ | Aggregated feature representation across all views |
| $\mathbf{z}$ | Multilayer perceptron used for classification |
| $\hat{\mathbf{y}}$ | Final model prediction |

## B Related Works

The vast majority of work on point cloud-based learning has focused on three dimensional problems, such as mapping and interpreting sensor readings in computer vision or localization tasks [43, 44, 65, 33, 58]. Early methods such as PointNet [43] and its successor PointNet++ [44] introduced permutation-invariant models with the ability to extract local and global features from 3D point clouds. Newer methods such as Dynamic Graph Convolutional Neural Network (DGCNN) [58] leverage dynamic graph representations to better extract rich features from input point clouds, while PointMLP [33] and Point Transformer [65] use standard Multilayer Perceptrons and local self-attention mechanism as the building blocks for better performance in classification and regression tasks. We are primarily interested in biological domains which are much higher dimensional, ranging from dozens in proteomics datasets to thousands in transcriptomic datasets, we require a new approach that is not limited by architectural decisions and is computationally efficient when working with high-dimensional data. Despite the advancements in these existing methods, they are fundamentally designed for 3D point clouds and struggle with performance and scalability in a high-dimensional setting as they rely on spatial heuristics, because local neighborhoods become less meaningful in high-dimension. As opposed to this, HiPoNet is able to scale to high-dimensional data while also preserving the geometry of the dataset.

Single-cell analysis has greatly benefited from graph-based methods that learns global structure from high-dimensional data. These methods typically rely on a *single* graph construct to infer relationships between cellular states. Methods such as UMAP [36] and t-SNE [52] perform non-linear dimensionality reduction by constructing a neighborhood graph and embedding cells into a low-dimensional space. On the other hand, PHATE [38] is a dimensionality reduction method that captures both global and local nonlinear structure but only constructs a *single* graph from the data. While these methods have been useful in understanding various biological processes, they may fail to organize cells based on processes governed by subsets of dimensions with just a single connectivity structure. In contrast to this, HiPoNet has the ability to model distinct cellular processes by leveraging its multi-view framework, preserving important geometric properties.

## C Proof of Theorem 4.1

The following Theorem is a more detailed version of Theorem 4.1, which states that the heat equation respects with the 0-homology of the underlying point cloud.

**Theorem C.1** (Connected Components on Simplicial Complexes)**.** *Let $\mathcal{S}$ be a simplicial complex of order $K$ with $m$ connected components, and let $\Sigma_k$ be the set of $k$-simplices ($k \leq K$). We partition $\Sigma_k = \Sigma_{k,1} \sqcup \Sigma_{k,2} \sqcup \ldots \sqcup \Sigma_{k,m}$, where each $\Sigma_{k,i}$ corresponds to the $k$-simplices in a single connected component. Let $S$ be a subset of $k$-simplices, and assume that $S$ is the union of several connected components: $S = \Sigma_{k,i_1} \sqcup \ldots \sqcup \Sigma_{k,i_k}$. Assume the initial condition, $u_k(\cdot, 0)$, of the diffusion equation has support contained in $S$. Then, for any $k$-th order simplex $\sigma_k \notin S$ and for all $t > 0$, we have:*

$$u_k(\sigma_k, t) = 0. \tag{5}$$

*Proof.* The Hodge-Laplacian $\Delta_k$ acts locally, meaning its $i, j$-th entry will be zero unless the $i$-th $k$ complex is a neighbor of the $j$-th $k$-complex (which can happen either via $i = j$ or via upper/lower adjacent neighbors). As a result, if the initial condition $u(\cdot, 0)$ has support contained in $S$, the solution $u(\sigma_k, t)$ must remain confined to $S$ for all $t > 0$. To formalize this, define a function $\tilde{u}(\sigma_k, t)$ as follows:

$$\tilde{u}_k(\sigma_k, t) = \begin{cases} u_k(\sigma_k, t), & \text{if } \sigma \in S, \\ 0, & \text{if } \sigma \notin S. \end{cases}$$

We will show that $\tilde{u}_k(\sigma_k, t)$ satisfies the same heat equation as $u(\sigma_k, t)$.

First, consider a simplex $\sigma_k \in S$. By definition, $\tilde{u}_k(\sigma_k, t) = u(\sigma_k, t)$ for all $t \geq 0$, and since $u_k(\sigma_k, t)$ satisfies the heat equation, it follows that:

$$\frac{\partial \tilde{u}(\sigma_k, t)}{\partial t} = \frac{\partial u(\sigma_k, t)}{\partial t} = -\Delta_k u_k(\sigma_k, t) = -\Delta_k \tilde{u}_k(\sigma_k, t),$$

where in the final equality we used the fact that $\Delta_k$ acts locally.

Next, consider a simplex $\sigma_k \notin S$. By definition, $\tilde{u}_k(\sigma_k, t) = 0$ for all $t \geq 0$, which implies $\frac{\partial \tilde{u}_k(\sigma_k, t)}{\partial t} = 0$. Furthermore, since $\Delta_k$ acts locally and $\sigma_k$ is not in $S$, we have also: $\Delta_k \tilde{u}_k(\sigma_k, t) = 0$ (since neighbors must be in the same connected component). Thus, $\frac{\partial \tilde{u}_k(\sigma_k, t)}{\partial t} = \Delta_k \tilde{u}_k(\sigma_k, t) = 0$, and so $\tilde{u}_k(\sigma_k, t)$ satisfies the heat equation.

Lastly, since $\tilde{u}_k(\sigma_k, t)$ satisfies the heat equation for all simplices $\sigma_k \in \mathcal{S}$ and coincides with the initial condition $u(\cdot, 0)$ on $S$, it follows from the uniqueness of solutions to the heat equation that $\tilde{u}_k(\sigma_k, t) = u_k(\sigma_k, t)$ for all $t > 0$. Therefore, for any simplex $\sigma_k \notin S$ and for all $t > 0$, we have: $u_k(\sigma_k, t) = \tilde{u}_k(\sigma_k, t) = 0$. $\qquad\square$

## D Proof of Theorem 4.3

Below, we will state Theorem D.1, which is a more detailed statement of Theorem 4.3. First, we will introduce some notation and preliminaries.

We will assume that the vertices of $\mathcal{G}(\mathcal{S})$, (i.e., the simplices $\sigma \in \mathcal{S}$) are ordered in a manner consistent with the size of the simplices, that is, all of the $k$ simplices come before all of the $k + 1$ simplices in our ordering. Let $N_k$ denote the number of simplicial complexes of order $k$ and let

$N = \sum_{k=0}^{K} N_k$ denote the cardinality of $\mathcal{S}$, i.e., the total number of simplices. For each $t \geq 0$, let $u_{\mathcal{G}}(\cdot, t) \in \mathbb{R}^N$ denote the solution to the graph heat equation:

$$\frac{\partial u_{\mathcal{G}}(\sigma, t)}{\partial t} = -\Delta_{\mathcal{G}} u_{\mathcal{G}}(\sigma, t), \tag{6}$$

where $\Delta_{\mathcal{G}} = B(\mathcal{G}(\mathcal{S}))B(\mathcal{G}(\mathcal{S}))^T$ is the graph Laplacian of $\mathcal{G}(\mathcal{S})$, and $B(\mathcal{G}(\mathcal{S}))$ is the incidence matrix of $\mathcal{G}(\mathcal{S})$. (If $\mathcal{S}$ is an oriented simplex, then $B(\mathcal{G}(\mathcal{S}))$ is the signed incidence matrix, otherwise it is the unsigned incidence matrix.) Let $u_k$ denote the solution to the heat equation associated to $\Delta_K$ in Eqn. 2 and, for $t \geq 0$ define the solution to the simplicial complex heat equation, $u_{\mathcal{S}}(\cdot, t) \in \mathbb{R}^N$ by

$$u_{\mathcal{S}}(\cdot, t) = [u_0(\cdot, t), \ldots, \ldots, u_K((\cdot, t)] \tag{7}$$

(i.e., the denote the concatenation of all of the $u_k$.)

**Theorem D.1** (Agreement of Heat Equation Solutions). *Let $\mathcal{S}$ be a simplicial complex, and let $\mathcal{G}(\mathcal{S})$ be its associated simplicial graph as defined in Definition 4.2. Let $u_{\mathcal{S}}(\sigma, t)$ denote the solution to the heat equation on $\mathcal{S}$. (See Eqn 7 and also Eqn. 2.) Similarly, let $u_{\mathcal{G}}(\sigma, t)$ denote the solution to the heat equation on $\mathcal{G}(\mathcal{S})$.*

*Assume that the initial conditions $u_{\mathcal{S}}(\cdot, 0)$ and $u_{\mathcal{G}}(\cdot, 0)$ are consistent, i.e.,*

$$u_{\mathcal{S}}(\sigma, 0) = u_{\mathcal{G}}(\sigma, 0) \quad \text{for all } \sigma \in V(\mathcal{G}).$$

*Then, for all $t > 0$ and for all simplices $\sigma \in \mathcal{S}$, the solutions agree, i.e., we have*

$$u_{\mathcal{S}}(\sigma, t) = u_{\mathcal{G}}(\sigma, t).$$

*Proof.* Since the edge set of $\mathcal{G}(\mathcal{S})$ is defined in terms of upper and lower adjacent neighbors, $\Delta_{\mathcal{G}}$ can be written in block diagonal form as

$$\Delta_{\mathcal{G}} = \begin{bmatrix} \Delta_0 & 0 & 0 & \cdots & 0 & 0 \\ 0 & \Delta_1 & 0 & \cdots & 0 & 0 \\ 0 & 0 & \Delta_2 & \cdots & 0 & 0 \\ \vdots & \vdots & \vdots & \ddots & \vdots & \vdots \\ 0 & 0 & 0 & \cdots & \Delta_{K-1} & 0 \\ 0 & 0 & 0 & \cdots & 0 & \Delta_K \end{bmatrix}. \tag{8}$$

This allows us to see that $u_{\mathcal{S}}$ also satisfies the graph equation equation since

$$-\Delta_{\mathcal{G}} \, u_{\mathcal{S}}(\cdot, t) = \begin{bmatrix} -\Delta_0 & 0 & 0 & \cdots & 0 & 0 \\ 0 & -\Delta_1 & 0 & \cdots & 0 & 0 \\ 0 & 0 & -\Delta_2 & \cdots & 0 & 0 \\ \vdots & \vdots & \vdots & \ddots & \vdots & \vdots \\ 0 & 0 & 0 & \cdots & -\Delta_{K-1} & 0 \\ 0 & 0 & 0 & \cdots & 0 & -\Delta_K \end{bmatrix} u_{\mathcal{S}}(\cdot, t)$$

$$= \begin{bmatrix} -\Delta_0 & 0 & 0 & \cdots & 0 & 0 \\ 0 & -\Delta_1 & 0 & \cdots & 0 & 0 \\ 0 & 0 & -\Delta_2 & \cdots & 0 & 0 \\ \vdots & \vdots & \vdots & \ddots & \vdots & \vdots \\ 0 & 0 & 0 & \cdots & -\Delta_{K-1} & 0 \\ 0 & 0 & 0 & \cdots & 0 & -\Delta_K \end{bmatrix} \begin{bmatrix} u_0(\cdot, t) \\ u_1(\cdot, t) \\ u_2(\cdot, t) \\ \vdots \\ u_K(\cdot, t) \end{bmatrix}$$

$$= \begin{bmatrix} -\Delta_0 \, u_0(\cdot, t) \\ -\Delta_1 \, u_1(\cdot, t) \\ -\Delta_2 \, u_2(\cdot, t) \\ \vdots \\ -\Delta_K \, u_K(\cdot, t) \end{bmatrix} = \begin{bmatrix} \frac{\partial u_0(\cdot, t)}{\partial t} \\ \frac{\partial u_1(\cdot, t)}{\partial t} \\ \frac{\partial u_2(\cdot, t)}{\partial t} \\ \vdots \\ \frac{\partial u_K(\cdot, t)}{\partial t} \end{bmatrix} = \frac{\partial}{\partial t} u_{\mathcal{S}}(\cdot, t).$$

Therefore, $u_{\mathcal{G}} = u_{\mathcal{S}}$ by uniqueness of solutions.

$\square$

# E The proof of Theorem 4.4

The following is a more detailed statement of Theorem 4.4, which states that the diffusion on simplices captures the geodesic distances.

**Theorem E.1** (Capturing Geodesic Distance). *Assume that a transformed point cloud $\tilde{\mathcal{X}}^{(v)}$ lies upon a Riemannian manifold $\mathcal{M}$. Let $\mathcal{S}$ be an oriented simplicial complex constructed from $\tilde{\mathcal{X}}^{(v)}$, and let $\mathcal{G}(\mathcal{S})$ be its associated simplicial graph. Denote the 0-th order heat kernel on $\mathcal{S}$ by $H_0^t(\sigma, \tau)$, where $\sigma, \tau$ represent simplices of any order in $\mathcal{S}$. So that $H_0^t u_0(\cdot, 0)$ solves the heat equation with initial condition $u_0(\cdot, 0)$. Then, we may approximate the geodesic distance $d_{\mathcal{M}}(\sigma, \tau)$ on the manifold $\mathcal{M}$ from $H_0^t$.*

*Proof.* Let $h_t$ denote the heat kernel on the underlying manifold. Varadhan's formula [53] shows that

$$\lim_{t \to 0} -4t \log h_t(\sigma, \tau) = d(\sigma, \tau)^2$$

Theorem 3 of [15] shows that the graph heat kernel $H_0^t$ uniformly converges to the manifold heat kernel $h_t$ and so the result follows from Theorem D.1 which establishes the equivalence of the graph heat equation and the heat equation on simplicial complexes. □

# F Proof of Corollary 4.5

**Corollary 5.** *The equivalence between the heat kernel on $\mathcal{S}$ and $\mathcal{G}(\mathcal{S})$ implies the equivalence of dimension, volume, and total scalar curvature on the respective underlying data manifolds.*

*Proof.* Let $h_t$ denote the heat kernel on the manifold $\mathcal{M}$. For any two points $x, y \in \mathcal{M}$. The eigenvalues $\lambda_i$ and corresponding eigenfunctions $\phi_i$ of the Laplace-Beltrami operator determine the manifold heat kernel via the spectral expansion:

$$h_t(x, y) = \sum_{i=0}^{\infty} e^{-\lambda_i t} \phi_i(x) \phi_i(y)$$

The eigenvalues are influenced by the volume and curvature of the manifold via Weyl's law [60] and the eigenvalue comparison theorems [11]. Thus, as in the proof of Theorem 4.4, the result follows from Theorem 3 of [15]. □

# G Data Cohorts

- **Melanoma**: We use data originally collected by Ptacek et al. [42] and made available in Vesely and Johnson [56], which consists of 54 melanoma patients, who underwent checkpoint blockade immunotherapy. The data contains approximately 489 to 1784 cells from the tumor micro-environments of each patient, resulting in a total of 11,862 cells. Each cell is characterized by 29 protein markers measured by Multiplexed Ion Beam Imaging (MIBI) [64, 3]. This data can be modeled as 54 point clouds in a 29-dimensional space, with sample sizes ranging from 489 to 1784 points per point cloud. The learning task is binary classification of whether the patient experienced recurrence or not (including stable disease as a 'non-recurrence'), predicted from the protein expression levels. We note that MIBI is able to measure the spatial locations of the cells as well as their protein counts. However, in our experiments, we do not use the spatial locations of MIBI. We make this choice so that, with this data set, we can assess the ability of methods to learn purely from the high-dimensional point clouds corresponding to protein counts.

- **Patient-derived Organoids (PDOs)**: We next consider data from Ramos Zapatero et al. [45] featuring patient-derived organoids (PDOs) from 12 different patients with colorectal cancer. Patient-derived organoids (PDOs) are cell cultures grown from patient tumor samples. Each culture, representing a PDO sample, is characterized by 44 proteins and contains approximately 1137 cells. Collectively, we consider 1,678 such cultures, each of which is viewed as a 44-dimensional point cloud, resulting in a total of 2 million cells. The goal is to infer treatment administered to the PDO based on cellular states, serving as a synthetic task for our model.

| Data | Task | K |
|------|------|---|
| Melanoma | Classification | 1 |
| | Persistence prediction | 1 |
| PDO | Classification | 1 |
| | Persistence prediction | 1 |
| DFCI | Outcome | 1 |
| Charville | Outcome | 2 |
| | Recurrence | 1 |
| UPMC | Outcome | 2 |
| | Recurrence | 1 |

Table A2: Order of simplicial complex

| Model | Accuracy |
|-------|----------|
| GCN | $41.67 \pm 25.00$ |
| GIN | $38.89 \pm 7.86$ |
| Graph Transformer | $50.00 \pm 16.67$ |
| HiPoNet | $63.60 \pm 0.00$ |

Table A3: Comparison of model accuracy on Spatial Data.

- **Spatial Transcriptomics (ST)**: We next consider Spatial Transcriptomics data generated using a 40-plex CODEX (CO-Detection by Indexing) immunofluorescent imaging workflow from Wu et al. [61], capturing 40 protein markers per cell across datasets from UPMC, DFCI, and Charville, comprising 500 patients, each of whom either had head-and-neck cancer or colorectal cancer. Here we construct views in two different manners:

  – *Spatial View:* Cellular coordinates describing how cells are arranged in two-dimensional tissue sections.
  – *Gene View:* Dozens of markers or transcripts measured per cell, detailing complex biological states.

  The overarching goal is to predict outcome of chemotherapy on cancer patients. Another goal is to predict cancer relaspe of recovered patients. This task particularly leverages HiPoNet's ability to fuse spatial context with transcriptional signals for clinically relevant insights in cancer research.

## H    Experimental Setup

The parameters of HiPoNet are optimized using the AdamW optimizer [32] with a constant learning rate of $10^{-4}$ and a weight decay of $10^{-4}$. We train the model for 100 epochs on every dataset for each model. We record the best metric (accuracy in classification tasks; mean squared error in regression tasks) achieved on the test set in each training run. We compare model performances on each dataset and task by the mean and standard deviation of their resulting five metric scores. For the spatial transcriptomics data, we use the folds from Wu et al. [61]. In Table A2, we describe order of the simplicial complex that we have used for each data set. Typically, our setup necessitates approximately two to three hours of training time for each dataset on a single NVIDIA A100 GPU.

### H.1    Baselines

We compare the performance of HiPoNet on the classification tasks outlined in Section 5 to two groups of alternative methods. First, to demonstrate the expressive power of our multiview graph embeddings over traditional graph neural network techniques, we construct K-nearest neighbor graphs of the input point cloud data and attempt to learn a classifier using several popular graph neural network architectures. These include Graph Convolutional Networks (GCN) [30], GraphSAGE [25], Graph Attention (GAT) Networks [54], Graph Isomorphism Networks (GIN) [63], and Graph Transformer Networks (GTNs) [16]. We discuss their performance on our high-dimensional tasks in Section 5.

| Num. of Views | Melanoma | PDO |
|---|---|---|
| 1 | $27.27 \pm 12.85$ | $59.80 \pm 1.99$ |
| 2 | $54.54 \pm 11.13$ | $61.10 \pm 0.17$ |
| 4 | $\mathbf{90.90 \pm 4.92}$ | $\mathbf{77.38 \pm 0.94}$ |

Table A4: Accuracies (mean $\pm$ standard deviation) from ablation on the number of graphs. Best result is bolded and second best is underlined.

| Model | Melanoma | PDO |
|---|---|---|
| HiPoNet | $\mathbf{90.90 \pm 4.92}$ | $\mathbf{77.38 \pm 0.94}$ |
| w/o multi-view | $27.27 \pm 12.85$ | $59.80 \pm 1.99$ |
| w/o structure | $46.08 \pm 8.38$ | $14.09 \pm 1.49$ |
| w/o reweighing | $56.36 \pm 7.60$ | $48.30 \pm 3.99$ |
| w/o wavelets | $70.90 \pm 14.93$ | $41.53 \pm 1.84$ |

Table A5: Ablation over components.

Second, we compare our method directly state-of-the art, bespoke point cloud learning methods. While DGCNN [58], PointNet++ [44], and PointTransformer [65] all perform very well on the three-dimensional problems they were designed to tackle, as shown in Table 4, they struggle to successfully classify high-dimensional point clouds.

## I  Computational Complexity

The computational complexity of one step of diffusion process, performed via sparse multiplication, has the complexity of $\mathcal{O}(\Sigma_{k=1}^{K} D_k N_k)$, where $D_k$ is a dimension of the features of $k$-simplices. To construct the wavelets, we need to perform $J$ steps of diffusion for $V$ views. Therefore, the cost of the SWT is $\mathcal{O}(VJ(\Sigma_{k=1}^{K} D_k N_k))$. For the second order scattering coefficients there are $\mathcal{O}(J^2)$ different choices of $j$ and $j'$. Thus the cost is $\mathcal{O}(VJ^2(\Sigma_{k=1}^{K} D_k N_k))$ The complexity of reweighing through Hadamard product is $\mathcal{O}(D_0 N_0)$. The complexity of VR-filtration is $\mathcal{O}(N_0^2)$. So, the total computational complexity of HiPoNet is $\mathcal{O}(N_0^2 + VJ^2(\Sigma_{k=1}^{K} D_k N_k))$.

## J  Ablation

In Table A4, we present an ablation study of HiPoNet showing the impact of varying number of views. The results indicate that four views is most effective for capturing the topology of the point clouds.

To assess the contribution of each component—multi-view learning, simplicial complex modeling, and wavelet transform—we conducted an ablation study on the **Melanoma** and **PDO** datasets. As shown in Table A5, removing multi-view learning causes the largest performance drop ($63.6\%$ on Melanoma, and $17.6\%$ on PDO), highlighting its central role. Additionally, we see that not using the structure of the data, but instead passing the point cloud features directly into an MLP classifier (without forming a simplicial complex), leads to severe degradation in performance on both tasks, with a $63\%$ drop on PDO and $44.8\%$ drop on Melanoma. Finally, we see that removing the wavelet transform (i.e., using diffusion only) lowers performance across both datasets (with a $20\%$ drop on Melanoma and a $25.85\%$ drop on PDO). These results confirm that each module plays a crucial role, with multi-view learning and structural modeling being especially vital.

To assess the sensitivity to the threshold, we conducted a sensitivity analysis on the Vietoris–Rips threshold, which governs the construction of higher-order simplices, using the single-cell classification datasets. As shown in Table A6, performance varies with the threshold. On **Melanoma**, the best result is achieved at 0.50, while on **PDO**, a lower threshold of 0.15 yields the highest accuracy, indicating that the optimal choice can be dataset-specific. To select the appropriate threshold, we tune it until reaching a critical point—beyond which the number of edges grows rapidly. These critical points are used in our analysis. Notably, we found that PDO has a critical point at 0.15, which leads to improved performance compared to what was reported in the main text. These results suggest that

| Threshold for V-Rips $\epsilon$ | Melanoma | PDO |
|---|---|---|
| 0.15 | $68.18 \pm 9.42$ | $\mathbf{77.38 \pm 0.94}$ |
| 0.25 | $65.45 \pm 21.70$ | $60.44 \pm 1.61$ |
| 0.50 | $\mathbf{90.90 \pm 4.92}$ | $63.10 \pm 0.00$ |
| 0.75 | $77.27 \pm 15.21$ | $68.92 \pm 0.94$ |
| 1.00 | $61.81 \pm 13.48$ | $64.96 \pm 0.85$ |
| 1.25 | $61.81 \pm 7.60$ | $62.00 \pm 0.51$ |

Table A6: Sensitivity analysis with respect to the Vietoris–Rips threshold.

| Kernel Bandwidth $\sigma$ | Melanoma | PDO |
|---|---|---|
| 0.25 | $61.81 \pm 7.60$ | $60.44 \pm 1.21$ |
| 0.50 | $61.81 \pm 11.85$ | $63.61 \pm 1.21$ |
| 0.75 | $65.45 \pm 7.60$ | $\mathbf{77.38 \pm 0.94}$ |
| 1.00 | $\mathbf{90.90} \pm 4.92$ | $62.44 \pm 1.46$ |
| 1.25 | $79.09 \pm 16.51$ | $62.00 \pm 0.51$ |

Table A7: Sensitivity analysis of kernel bandwidth (%).

the model is sensitive to this hyperparameter, and careful selection of the threshold is important for achieving strong performance.

To assess the effect of kernel bandwidth, we performed a sensitivity analysis by varying the bandwidth used during graph construction. As shown in Table A7, performance on the Melanoma dataset initially improves with increasing bandwidth and peaks at $1.00$ ($90.90 \pm 4.92$), after which it slightly declines. On the PDO dataset, however, the best performance is observed at $0.75$ ($77.38 \pm 0.94$), suggesting that overly large bandwidths may oversmooth the neighborhood structure in some domains. These results indicate that kernel bandwidth is a sensitive hyperparameter and should be tuned based on dataset characteristics. We will incorporate this analysis and its implications into the revised manuscript.

Table A8 presents results across four tasks—PDO, Charville (Outcome and Recurrence tasks), and UPMC (Recurrence task)—with $K = 1, 2, 3$ representing the maximum dimension of simplices used during complex construction. We observe that for *PDO*, performance peaks at $K = 1$, suggesting that lower-order interactions are sufficient. In contrast, for outcome prediction on *Charville* and *UPMC*, higher-order structures ($K = 2$) improve performance, particularly for the UPMC dataset where AUC increases from $0.538$ to $0.6044$. Notably, performance saturates or slightly drops at $K = 3$, indicating diminishing returns from overly complex topology.

# K    Broader Impacts

This work introduces HiPoNet, a neural network for high-dimensional point cloud analysis with applications in biological data such as single-cell and spatial transcriptomics. Positive societal impacts include advancing our understanding of complex biological systems, enabling discoveries in disease mechanisms, and contributing to the development of more effective treatments through improved modeling of gene and cell interactions. Potential negative impacts include the risk of over-reliance on model predictions in critical biological or clinical decision-making without sufficient experimental validation. Additionally, applying HiPoNet to sensitive biomedical datasets raises privacy concerns if appropriate safeguards are not maintained. We recommend that HiPoNet be used primarily for hypothesis generation rather than direct clinical application, with findings validated through rigorous experimental studies. Responsible data handling practices and collaboration with domain experts are essential to mitigate these risks.

# L    Limitations

While HiPoNet advances the modeling of high-dimensional point clouds, it has several limitations. First, although HiPoNet is relatively efficient, training on very large datasets still requires substantial compute resources, especially for larger values of $K$. Second, the current model has been primarily validated on biological datasets; its generalization to other high-dimensional domains remains to be

| Dataset / Task (Metric) | $K = 1$ | $K = 2$ | $K = 3$ |
|---|---|---|---|
| PDO (Accuracy) | **77.38 $\pm$ 0.94** | 72.30 $\pm$ 0.18 | 70.76 $\pm$ 0.71 |
| Charville (Outcome AUC-ROC) | 0.55 $\pm$ 0.001 | **0.598 $\pm$ 0.12** | 0.539 $\pm$ 0.05 |
| Charville (Recurrence AUC-ROC) | **0.681 $\pm$ 0.01** | **0.681 $\pm$ 0.01** | **0.681 $\pm$ 0.01** |
| UPMC (Recurrence AUC-ROC) | 0.538 $\pm$ 0.03 | **0.6044 $\pm$ 0.0** | 0.583 $\pm$ 0.01 |

Table A8: Performance across datasets and tasks for varying $K$.

explored. Finally, HiPoNet focuses on predictions rather than inferring causal relationships, limiting its direct applicability for causal inference tasks.

