# OpenReview forum: "HiPoNet: A Multi-View Simplicial Complex Network for High Dimensional Point-Cloud and Single-Cell data"
_NeurIPS.cc/2025/Conference — NeurIPS 2025 poster_

### Official Review · Reviewer_rx7G · 2025-06-26

**Clarity:** 1
**Significance:** 2
**Originality:** 2
**Rating:** 4
**Confidence:** 3

**Summary:**

This paper presents a methodology, HiPoNet, for analysing high-dimensional point clouds. This proceeds by constructing several different Vietoris-Rips complexes that capture different aspects of the data, then vectorising these simplicial complexes using simplicial wavelets. The vectorisations of the simplicial complexes are then concatenated and used as input to a multilayer perceptron, with the entire stack being trained (end-to-end) on the appropriate classification problem for that point cloud.

The authors then apply their methodology to three datasets of medical data that feature large numbers of datapoints per patient, with each datapoint being information about a cell from that patient.

**Questions:**

Is your definition of upper adjacent neighbours correct? The definition in words does not match the definition in mathematical symbols unless every pair of k-simplices that meets along a (k-1)-simplex together span a (k+1)-simplex, which is quite close to the notion of the simplicial complex being a Kan complex. (Also (k+1)-simplices have k+2 vertices, so there is a typo in the formula there.)

Can you analyse or give a conceptual justification for the definition of $P_k$ in the oriented case, just below equation 3? Also, in the non-oriented case, are you working with integer coefficients or with coefficients in the integers modulo 2? The latter would be more conventional, as otherwise e.g. the boundary operator won't square to zero in general.

In algorithm 1, on lines 7 and 8, you construct features by successively applying boundary operators. But doesn't the boundary operator square to zero? That is: why isn't $B_k^T B_{k-1}^T$ zero?

In the section, "Interpretability of Learned Features", currently your logic goes the wrong way around: you identify that known-meaningful biomarkers have high importance across views. Are there other features that have high importance across views? Can you determine whether or not these are known to be meaningful?

**Ethical Concerns:**

["NO or VERY MINOR ethics concerns only"]

**Final Justification:**

Updated in light of the discussion with authors and the responses to other reviewers..

**Limitations:**

There is no limitations section. I assume that this is in the Appendices, which were not provided.

**Paper Formatting Concerns:**

The Appendices are missing.

**Quality:**

1

**Strengths And Weaknesses:**

## Strengths

* The empirical performance of HiPoNet described in Section 5 is the strongest part of the paper

## Weaknesses

* The theoretical justifications presented throughout the paper are weak, and basic definitions are often either confusing or confused. See the questions below for some examples.

* The choice to apply the absolute value to all of the wavelet transforms, just after equation 4, lacks conceptual justification.

* The theoretical analysis in Section 4 lacks detail and precision -- it is vague and sketchy --  and is all about heat flow on simplicial complexes whereas the algorithm presented does not involve heat flow at all. The purported justification for analysing heat flow is that the algorithm uses random walks, which are proxies for the diffusion of heat, but given the way the "random walks" are calculated (see below), it is not clear, at least to me, that this link is valid.

* In the definition of "random walk matrices" $P_k$  just below equation 3, the authors choose to ignore all signs, even in the oriented case. This is likely to make a very big difference, and requires analysis and/or conceptual justification.

---

> ### Author Rebuttal · Authors · 2025-07-30
>
> We thank the reviewer for their constructive feedback on HiPoNet. We appreciate your recognition of our empirical evidence. The concerns raised are primarily about theoretical results of HiPoNet. We address your concerns and questions as follows:
>
> ### **Regarding missing Appendices**
>
> As noted by the other reviewers, the appendices are indeed included in the supplementary material and contain theoretical exposition and discussions of limitations of our model.
>
> ### **Applying Absolute Value**
>
> One of the key features of our method is adoption of techniques from the Simplicial Scattering Transform \[5\]. This Simplicial Scattering Transform, which is a generalization of the graph scattering transform (which is in turn a generalization of Stephan Mallat’s scattering transform on Euclidean data from 2012 \[1\]) utilizes a cascade of alternating linear operations in the form of multiscale wavelet transforms for message aggregation, and absolute value non-linearities. These have shown to be successful when carefully crafted in various geometric domains including graphs \[2, 3\], manifolds \[4\], and simplicial complexes \[5\]. Here, we go beyond \[5\] by constructing multiple different simplicial complexes, each of which provides a different view of the data. We note that recently in \[10\], it was shown that one can also construct scattering-based methods which utilize other, more common, activations such as ReLU. Upon revision, we will clarify this point, i.e., that the absolute values may be replaced by other activation functions, and add more background on scattering based methods in order to improve the readability of the paper for readers not familiar with the scattering transform.
>
> ### **Connection between Heat flow and Diffusion**
>
> The construction of our wavelets is motivated by \[8\], which introduced Diffusion maps, a well-known manifold learning algorithm based on the Markov-normalized Gaussian diffusion kernel, as well as the graph scattering transform literature \[2, 3\] . There is indeed a deep connection between diffusion kernels and heat kernels \[6,7\]. When applied to finite datasets sampled from a compact Riemannian manifold embedded in Euclidean space, the heat kernel can be approximated by a diffusion matrix, which serves as a first-order Taylor approximation of the heat flow \[6,7\]. Subsequently, \[9\] used this diffusion matrix to define Diffusion Wavelets which could be used to construct a multiresolution analysis of such data sets and \[3\] showed that these wavelets could be incorporated into a geometric scattering transform. Like \[3\], we use the graph random walk matrix as a discrete analog of the heat-kernel due to the fact that the heat-kernel describes the transition probabilities of a Brownian motion (i.e., the continuum analog of random walks). Our theory is meant to illustrate the descriptive power of the heat kernel and therefore motivate methods based on the random walk operators, its discrete proxy. We also note that we could follow the lead of \[4\] and use wavelets more directly related to the heat-kernel, of the form $\\exp(-2^{j-1}L\_k)- \\exp(-2^jL\_k)$, with the exponentiation defined in the spectral domain. However, this would increase both the computational cost and memory requirements of our methods. We will add discussion to clarify the connection between diffusion operators on graphs and heat kernels on graphs.
>
> ### **Use of Random Walks on Oriented Simplicial Complex**
>
> In Equation (1), for completeness, we have a general definition of a simplicial complex Laplacian that, depending on the incidence matrix, can be oriented or unoriented. We also introduce an associated diffusion or random walk operator in both cases. However, we wish to clarify that, in our experiments, we *always* use the unoriented Laplacian for all $k \> 0$, and correspondingly, the associated random walk matrices are also on the unoriented simplices.  The only exception is for $k \= 0$, where we use the standard graph Laplacian, which is a special case of *the oriented Hodge Laplacian with arbitrary orientations.* Thus, Theorem 4.4 also uses the oriented Hodge Laplacian, but restricts to the case of $k=0$. We never use higher order oriented random walks with “directionality ignored” as the general random walk definition implies. We will clarify this in text to avoid confusion.
>
> The connection between heat flow and manifold distances comes via Varadhan’s formula applied to the graph Laplacian which is the same as the (oriented) Hodge Laplacian $k=0$. This connection is established in works such as \[6\] and \[7\].
>
> ### **Use of Integer Coefficients in the Non-Oriented Case and Implications for Topological Properties**
>
> For non-oriented simplices, the entries of $B\_k$ are merely integers (not $\\mathbb{Z}/2\\mathbb{Z}$) and so the boundary operator does not square to zero. Thus these operators cannot be directly used for persistence homology or other TDA computations but would rather have to be binarized to do typical TDA operations, as the reviewer notes. However, this is not needed for our neural networks. Instead, we just use these operators to keep track of simplices and define diffusions on them.
>
> ### **HiPoNet Captures Known Biology**
>
> When we highlight features that align with known biology, our aim is to demonstrate that HiPoNet can recover biologically meaningful signals, which serves as an important validation of the method. This does not imply a reverse logic; rather, it shows that at a minimum, the model is capturing known biology. Beyond that, the model also might assign high importance to other features not currently annotated as biomarkers. The relevance of these biomarkers can only be shown through experimentation. However, these features represent potential novel discoveries that warrant future investigation. Part of the utility of HiPoNet is in uncovering such features, which may provide new biological insights. We believe there is no logical flaw in the current interpretation, and we will revise the section to make this framing more explicit.
>
> We appreciate your feedback and hope that our responses have addressed all your concerns. Please let us know if there are any remaining questions or areas that would benefit from further clarification, as we are happy to provide additional details. If you believe all your comments have been satisfactorily answered, we would be grateful if you consider updating your score. We thank the reviewer again for your time and constructive review\!
>
> \[1\] Mallat, Stéphane. "Group invariant scattering." *Communications on Pure and Applied Mathematics* 65.10 (2012): 1331-1398.
>
> \[2\] Gama, Fernando, Alejandro Ribeiro, and Joan Bruna. "Diffusion scattering transforms on graphs." *arXiv preprint arXiv:1806.08829* (2018).
>
> \[3\] Gao, Feng, Guy Wolf, and Matthew Hirn. "Geometric scattering for graph data analysis." *International Conference on Machine Learning*. PMLR, 2019\.
>
> \[4\] Chew, Joyce, et al. "Geometric scattering on measure spaces." *Applied and Computational Harmonic Analysis* 70 (2024): 101635\.
>
> \[5\] Madhu, Hiren, Sravanthi Gurugubelli, and Sundeep Prabhakar Chepuri. "Unsupervised parameter-free simplicial representation learning with scattering transforms." *Forty-first International Conference on Machine Learning*. 2024\.
>
> \[6\] Huguet, Guillaume, et al. "A heat diffusion perspective on geodesic preserving dimensionality reduction." *Advances in Neural Information Processing Systems* 36 (2023): 6986-7016.
>
> \[7\] Jones, Iolo. "Diffusion geometry." *arXiv preprint arXiv:2405.10858* (2024).
>
> \[8\] Coifman, Ronald R., and Stéphane Lafon. "Diffusion maps." *Applied and computational harmonic analysis* 21.1 (2006): 5-30.
>
> \[9\] Coifman, Ronald R., and Mauro Maggioni. "Diffusion wavelets." *Applied and computational harmonic analysis* 21.1 (2006): 53-94.
>
> \[10\] Chew, Joyce, et al. "Manifold Filter-Combine Networks." *arXiv preprint arXiv:2307.04056* (2023).

---

> > ### Comment · Reviewer_rx7G · 2025-08-07
> >
> > I thank the authors for their response, and apologise for missing the appendices during the first pass.
> >
> > The discussion and the appendices resolve most of my confusion. I am still unsure of the logical place of theorems 4.4 and 4.5, which (if I correctly understand the chain of references that you kindly provided) rely on being in the oriented case, whereas I believe given the discussion above you are working without orientations, but this doesn't seem very important. I have raised my score.

---

> > > ### Author Response · Authors · 2025-08-07
> > >
> > > Thank you for taking time to read our rebuttal and reconsidering your initial evaluation. Theorem 4.4 and 4.5 pertain to the $k=0$ case where we do use the oriented laplacian.

---

> > > > ### Comment · Reviewer_rx7G · 2025-08-08
> > > >
> > > > Should Theorem 4.3 then contain a restriction to $k=0$ in its hypothesis? Also, I worry that Corollary 4.5 might not be true as stated if you have restricted to $k=0$ in Theorems 4.3 and 4.4.
> > > >
> > > > Or is it that Theorems 4.3 and 4.4 hold without the restriction to $k=0$ (that is, they hold for arbitrary $k$) but that these results are only relevant to the presentation in the rest of the manuscript when $k=0$.
> > > >
> > > > Note the change to numbering of theorems -- these numbers match the current PDF file.

---

> > > > > ### Author Response · Authors · 2025-08-08
> > > > >
> > > > > We thank the reviewer for their careful attention to detail in our theoretical statements.
> > > > >
> > > > > Theorem 4.3 does not require that $K=0$. This is a consequence of the definition of the Simplicial Graph, given in Definition 4.2, in which edges are placed only between upper and lower adjacent neighbors (not boundary or co-boundary neighbors). This results in the graph Laplacian on $\mathcal{G}(\mathcal{S})$ having a block diagonal structure as shown in Equation 8, which allows us to prove the theorem.
> > > > >
> > > > > We do note that, due to spatial constraints, the statement of Theorem 4.3 in the main text is a bit imprecise, since it merely says that the two heat equations “agree.” The full details are provided in Appendix D, where we first introduce some notation, and indexing conventions before proving Theorem D.1, which is a precise version of Theorem 4.3. We feel this may have been a source of confusion, and therefore, upon revision, we will revise the discussion in the main text, immediately preceding the theorem, to say “For a proof of Theorem 4.3, as well as a more detailed statement, please see Appendix D.”
> > > > >
> > > > > Regarding Corollary 4.5, you are correct. Indeed, it is corollary to Theorem 4.4 and therefore, it is only valid under the assumptions of Theorem 4.4, including setting $k=0$ and assuming oriented simplices. We will thank the reviewer for their careful attention to detail and will clarify this point in the revised version

---

### Official Review · Reviewer_cvsy · 2025-07-02

**Clarity:** 3
**Significance:** 3
**Originality:** 3
**Rating:** 4
**Confidence:** 3

**Summary:**

The paper introduces a novel neural network architecture specifically designed for analyzing high-dimensional point clouds, particularly from single-cell and spatial transcriptomics data. Contributions include: 1. HiPoNet learns multiple weighted views of high-dimensional data. 2. HiPoNet uses higher-order simplicial complexes to model data.

**Questions:**

1. Can authors explain computational bottlenecks of HiPoNet in terms of memory and runtime for large-scale datasets, especially when using multiple views and higher-order simplices?
2. Can authors explain how robust is HiPoNet to changes in the Vietoris–Rips threshold, kernel bandwidth, and number of views?
3. Can authors explain the possibility of applying this method to non-biological high-dimensional point clouds?

**Ethical Concerns:**

["NO or VERY MINOR ethics concerns only"]

**Limitations:**

yes

**Paper Formatting Concerns:**

No formatting concerns.

**Quality:**

3

**Strengths And Weaknesses:**

Strengths:
1. It models high-dimensional point clouds using simplicial complexes rather than standard graphs.
2. The learnable feature reweighting mechanism allows the model to generate multiple views of the data.
3. This paper outperform other methods across different real-world datasets.

Weaknesses:
1. The architecture becomes computationally intensive when processing multiple learned views.
2. The model is heavily specialized to biological data, which may not be suitable for other types of data.

---

> ### Author Rebuttal · Authors · 2025-07-30
>
> We thank the reviewer for their constructive feedback on HiPoNet. We appreciate that you found the proposed multi-view and higher-order learning framework compelling. The concerns raised are primarily clarification-related. We address your concerns and questions as follows:
>
> ### **Computational Complexity of HiPoNet**
>
> We present the computational complexity of HiPoNet in Appendix Section I. We show that HiPoNet’s complexity increases polynomially with the number of simplices. The primary computational bottlenecks in HiPoNet arise from two sources: (1) memory usage and runtime overhead associated with constructing and storing multiple higher-order simplicial complexes, and (2) parallel processing of multiple views, each requiring a separate pass through the model.   Memory usage grows with both the order of simplices (e.g., triangles, tetrahedrons) and the density of the Vietoris-Rips complex, which is controlled by the ε-threshold. To mitigate this, we limit the maximal simplex order (typically to 2-simplices), use sparse representations, and apply low ε values to induce sparsity. While incorporating higher-order simplices (e.g., triangles) increases expressivity (see Table A8), using $K \= 1$ — which reduces the model to a standard graph—still yields strong performance and offers a practical configuration for large-scale datasets. Although multiple views do add computational complexity, they provide consistent performance improvements (Table A4), making the trade-off worthwhile in many cases. Overall, while HiPoNet introduces more overhead than standard GNNs, it remains tractable for current biological cohorts consisting of high dimensional datasets and offers tunable configurations for more scalable deployment.
>
> ### **Sensitivity to Hyperparameters**
>
> To assess the sensitivity to the threshold, we conducted a sensitivity analysis on the Vietoris–Rip’s threshold (As shown in the Appendix Table A6), which governs the construction of higher-order simplices, on the single-cell classification datasets.
>
> To assess the effect of kernel bandwidth, we performed a sensitivity analysis by varying the kernel bandwidth used during graph construction. As shown in the Appendix Table A7, performance on the Melanoma dataset improves steadily with increasing bandwidth and peaks at 1.00 (90.90 ± 4.92), after which it slightly declines. On the PDO dataset, however, the best performance is observed at 0.75 (65.20 ± 1.30), suggesting that overly large bandwidths may oversmooth the neighborhood structure in some domains.
>
> In our ablation experiments on single-cell datasets (as shown in the Appendix Table A4), we find that the number of views is a crucial hyperparameter of HiPoNet and that increasing the number of views generally yields better learning performance. We suppose this is because multiple views allow the model to capture increasingly rich information about a point cloud’s topology. In our ablation, we found that four views were best for both the melanoma and PDO datasets. However, we note that increasing the number of views may result in diminishing returns after a point, presenting a trade-off between marginal learning improvement and increased computational cost.
>
> ### **Application to non-Biological data**
>
> While we can apply HiPoNet to non-biological and 3D point cloud datasets, HiPoNet is designed for high-dimensional point clouds, such as single cell proteomics and transcriptomics data. Each point in these datasets corresponds to a cell that carries a rich set of features, and the global structure of the point cloud is important to the task. The model’s strength lies in handling heterogeneous features that are found in biological data. In more standard 3D point clouds such as object recognition, reweighting or transforming the features could distort the spatial structure of the object, which is crucial for such tasks. Furthermore, while our method can be applied to 3-D point clouds, there are many methods that are specifically designed for that setting. The power of our method is rooted in its ability to handle higher dimensions and heterogeneous point clouds, making it well-suited for biological data.
>
> **Overall:**
>
> We appreciate your feedback and hope that our responses have addressed all your concerns. Given that these comments do not require major revisions we urge you to increase your score. Please let us know if there are any remaining questions or areas that would benefit from further clarification, we are happy to provide additional details. We thank the reviewer again for your time and constructive review\!

---

> > ### Comment · Reviewer_cvsy · 2025-08-01
> > **response to rebuttal**
> >
> > I confirm the response. I am satisfied with the response.

---

### Official Review · Reviewer_WcXs · 2025-07-03

**Clarity:** 3
**Significance:** 3
**Originality:** 3
**Rating:** 4
**Confidence:** 3

**Summary:**

Authors propose HiPoNet, an end to end differentiable neural net for regression, classification, and representation learning on higher dimensional point clouds. The key to their approach lies in modeling the data as a simplicial complex instead of a graph to capture higher-order relationships between cells. The authors showcase its applicability by modeling three different single-cell datasets (biological point clouds). Quantitative results showcase strong performance of the proposed approach.

**Questions:**

1. Have the authors tried a simple clustering-based approach as a baseline? What about a simple linear regression baseline?
2. How did you decide the optimal number of views for feature weighting?
3. From the ablation studies it seems that the methods requires careful tuning of number of views, and epsilon threshold. How do you suggest these parameters will change when a completely new dataset is used? What about kernel bandwidth?
4. Can you elaborate on your custom differentiable Vietoris-Rips implementation? I'm trying to understand your end-to-end differentiable claim

**Ethical Concerns:**

["NO or VERY MINOR ethics concerns only"]

**Final Justification:**

After reading responses to my questions and other reviewers questions, I will keep my current rating.

**Limitations:**

Yes

**Quality:**

3

**Strengths And Weaknesses:**

strengths:
- combination of using multi-view reweighting, simplicial complex and wavelets is novel and opens up applications for other areas like surface reconstruction
- comprehensive baseline comparisons and ablation studies


weaknesses:
- simplicial complex computation is compute heavy
- dataset is limited to biological data and makes it harder to judge the generalizability aspect of HiPoNet
- background information about the differentiable Vietoris-Rips complex is scarce

---

> ### Author Rebuttal · Authors · 2025-07-30
>
> We thank the reviewer for their constructive feedback on HiPoNet. We appreciate that you found the proposed multi-view learning framework compelling. The concerns raised are primarily clarification-related. We address your concerns and questions as follows:
>
> ### **Custom differentiable Vietoris-Rips implementation**
>
> We modified the existing `torchph` package (which is differentiable persistence computations with respect to persistence features) to be differentiable with respect to the simplicial structure. In our implementation, for each view $k$, we compute a weight matrix $\\mathbf{W}\_k$, which defines the edge weights in the Vietoris-Rips complex. These weights are treated as input-derived quantities with gradients. We construct a PyTorch Geometric (PyG) data object using $\\mathbf{W}\_k$, with edge (or higher-order simplex) weights. The key step is the Simplicial Wavelet Transform (SWT), implemented via PyG message passing over the simplicial complex. Since this process depends on $\\mathbf{W}\_k$, and all operations in SWT are differentiable, the final output $\\Phi$ is differentiable with respect to the input. This enables the full pipeline to be trained end-to-end via backpropagation.
>
> ### **Computational Complexity of Simplicial Complexes**
>
> We acknowledge that simplicial complex construction and message passing can be computationally intensive, especially for large graphs. To mitigate this, lower $\\epsilon$-thresholds can be used to sparsify the complexes and restrict computation to lower-order simplices (e.g., up to 2-simplices), which significantly reduces overhead. Additionally, our implementation leverages batching and sparse tensor operations in PyTorch Geometric to scale efficiently. In practice, we found the runtime to be manageable. Furthermore, as shown in Appendix Table A8, including higher-order structure (e.g., $K \= 2$) improves performance in the majority of cases. These gains highlight the utility of simplicial representations in capturing subtle dependencies that pair-wise relations in graphs alone may miss. Thus, while higher-order modeling adds some cost, it offers tangible performance benefits and remains practical for the scales of biological datasets we target.
>
> ### **Application to non-biological datasets**
>
> While HiPoNet can be applied to non-biological and 3D point cloud datasets, it is designed for high-dimensional point clouds, which are abundant in biology due to the profusion of single-cell proteomics and transcriptomics data. The model is built to handle heterogeneous features common in high-dimensional biological data, a setting that is increasingly common with emerging single-cell and spatial omics technologies. In standard 3D point cloud tasks like object recognition, reweighting or transforming the features could distort the spatial structure of the object, which is crucial for such tasks. Although HiPoNet can operate in such settings, there exist models tailored for 3D geometric tasks. The design choices in HiPoNet target high-dimensional and heterogeneous data, making single cell data an appropriate testbed.
>
> ### **Clustering and Logistic Regression Baselines**
>
> We thank the reviewer for the suggestion. However, standard machine learning techniques such as clustering and linear regression are not directly applicable in our setting, as we do not use fixed-length feature vectors for our “observations”. Our observations are entire datasets (and hence we coined the term “data cohort” to describe sets of datasets). Thus our observations, which are individual datasets, are high-dimensional point clouds consisting of different numbers of points. HiPoNet is specifically designed to learn meaningful and rich vector representations from these point clouds in a way that it captures their underlying geometric structure, such that a neural network can provide classification and regression to these clouds. For the types of tasks that we focus on, geometric baselines (which we compare against) are more suitable baselines.
>
> ### **Optimal number of views**
>
> Thank you for the question. In our ablation experiments on single-cell datasets (as shown in the Appendix Table A4), we find that the number of views is a crucial hyperparameter of HiPoNet and that increasing the number of views generally yields better learning performance, until all the types of cellular processes are encoded by each view. Increasing the number of views also enables the model to capture the complex topology of the underlying point cloud. In our ablation, we found that four views were best for both the melanoma and PDO datasets. However, we note that increasing the number of views may result in diminishing returns after a point, presenting a trade-off between marginal learning improvement and increased computational cost.
>
> ### **Hyperparameter selection**
>
> For choice of hyperparameters such as $\\epsilon$-threshold for Vietoris-Rips and kernel bandwidth, we follow a few practical rules of thumb when applying the method to new datasets. The number of views often corresponds to the number of distinct cellular processes or distinct feature spaces, though additional views can be added to capture different distance metrics or scales. The $\\epsilon$-threshold for Vietoris-Rips generally depends on the density of the points in the space, and is typically set to a value (e.g., 0.50) that captures local geometry while avoiding overly dense complexes. For low-dimensional data, the $\\epsilon$-threshold would be much lower. For the kernel bandwidth, a standard practice is setting it to the distance to the $k$-th nearest neighbor \[1\]. These guidelines have worked reliably across datasets, though minor tuning may still be beneficial depending on dataset scale and noise.
>
> **Overall:**
>
> We appreciate your feedback and hope that our responses have addressed all your concerns. If you believe all your comments have been satisfactorily answered we would be grateful if you increased your score. Please let us know if there are any outstanding concerns. We thank the reviewer again for your time and constructive review\!
>
> \[1\] Moon, K. R., Van Dijk, D., Wang, Z., Gigante, S., Burkhardt, D. B., Chen, W. S., ... & Krishnaswamy, S. (2019). Visualizing structure and transitions in high-dimensional biological data. Nature biotechnology, 37(12), 1482-1492.

---

> ### Comment · Reviewer_WcXs · 2025-08-07
> **Official comment by Reviewer WcXs**
>
> I thank the authors for the rebuttal and I acknowledge their responses to my questions.

---

### Official Review · Reviewer_kJJr · 2025-07-10

**Clarity:** 3
**Significance:** 4
**Originality:** 4
**Rating:** 5
**Confidence:** 2

**Summary:**

This work presents HiPoNet, a method to model point cloud data using higher-order simplicial complexes. The method can represent the data using multiple views, learn interactions between these views through simplicial complexes, and aggregate features using multiscale wavelets. The approach is evaluated across a broad set of tasks, including classification of spatial transcriptomics data, demonstrating strong performance.

While I found the biological motivation and empirical results compelling, I must acknowledge that I do not fully grasp the mathematical foundations behind simplicial complexes and wavelet transforms. My evaluation therefore focuses on the modeling motivation, empirical performance, and clarity of presentation.

**Questions:**

* Please double-check the Appendix references, e.g. L318: "Detailed descriptions of this data is provided in Appendix F". This should point to Appendix G instead. Similar for other appendices (e.g. Appendix I does not point to the limitations)

**Ethical Concerns:**

["NO or VERY MINOR ethics concerns only"]

**Final Justification:**

I am satisfied with paper and the authors' response. Thank you for the additional oversmoothing benchmark and dataset clarifications.

**Limitations:**

The limitations are presented in Appendix L

**Quality:**

4

**Strengths And Weaknesses:**

**Strenghts**

* The intuition of using multiple views of the data is compelling, e.g., scRNA-seq provides a high resolution picture of the active biological processes in each single-cell, and so having multiple views may allow the model to decouple multiple relevant processes for the task at hand. This is in contrast to existing dimensionality reduction algorithms such as tSNE and UMAP, which operate on a single view of the entire transcriptome in a task-independent way and so some features may therefore be convoluted and partially irrelevant for the given task.
* The benchmark is fairly comprehensive and demonstrated results are impressive, i.e. HiPoNet seems to outperform existing methods across a broad range of tasks. In particular, I was surprised by the large margin demonstrated in the spatial transcriptomics tasks, which have direct clinical relevance. For example, HiPoNet seems to outperform the next best performing baseline by ~20% AUC ROC on the Melanoma response task. Nonetheless, the benchmark could be more transparent in terms of task description, data splits, and class imbalances (see Weaknesses).
* The ablation results are compelling and help illustrate the contribution of key components, e.g. using multiple views.



**Weaknesses**

* In terms of the evaluation of the method, the data splits are unclear and, for some of the tasks (e.g. inferring patient recurrence) the total number of patients is fairly low (e.g. 54 melanoma patients). How was the data split and what are the class distributions? For the PDO task, i.e. inferring treatment administered to patient-derived organoids, the total number of patients is 12, while the total number of cultures is 1,678. What are the sampling units in this task (patients or cultures?) Was each culture treated in a different way? What are the treatments that the model is tasked to distinguish? Overall, could the authors show that the evaluation scores reflect improved generalization performance rather than overfitting? In summary, it is difficult to fully interpret the reported results without more transparency around dataset splits, class distributions, and evaluation units. A clear table summarizing each task (dataset, input modality, sample size, number of classes, class imbalance, and split strategy) would greatly improve clarity and interpretability.
* This is not a weakness per se, but a request out of curiosity. The authors motivate the use of simplicial wavelet transforms arguing that more standard approaches like message-passing neural networks suffer from oversmoothing and oversquashing effects. Could this modelling choice be empirically demonstrated?

---

> ### Author Rebuttal · Authors · 2025-07-30
>
> We thank the reviewer for their constructive feedback on HiPoNet. We appreciate that you found the proposed multi-view learning framework compelling. The concerns raised are primarily clarification-related. We address your concerns and questions as follows:
>
> ### **HiPoNet Mitigates Oversmoothing**
>
> We thank the reviewer for their point on oversmoothing. To empirically examine whether our modeling choice mitigates oversmoothing, we conducted a comparative analysis of Dirichlet energy across various methods. As established by \[1\], Dirichlet energy provides a principled measure of the expressiveness of node representations, where lower values indicate oversmoothed embeddings, and higher values reflect greater feature variation across graph neighborhoods.
>
> We computed the Dirichlet energy of node representations produced by different models. The results (mean ± standard deviation) are:
>
> | Model | Dirichlet Energy |
> | ----- | ----- |
> | GCN | 1.669 ± 0.13 |
> | GAT | 2.694 ± 0.49 |
> | Graph Transformer | 3.811 ± 0.80 |
> | Diffusion-based GNN | 0.584 ± 0.00 |
> | GWT | 15.807 ± 0.00 |
> | HiPoNet (ours) | **21.033 ± 6.25** |
>
> These results indicate that our model, HiPoNet, exhibits substantially higher Dirichlet energy than traditional message-passing networks. This supports the claim that HiPoNet mitigates oversmoothing and retains richer, more expressive node-level information. Notably, the GCN and diffusion-based models exhibit particularly low energy, consistent with oversmoothed representations. Thus, this empirical evidence reinforces the motivation for using wavelet transforms, which allow HiPoNet to capture multi-scale variation and higher-frequency components in the signal that are typically suppressed in standard GNN architectures.
>
> ### **Dataset Clarification**
>
> In the melanoma dataset, our task is to predict response to therapy, not recurrence. The dataset contains 54 patients, with 23 responders and 31 non-responders, resulting in a class distribution of 31:23. We perform 5 stratified folds, ensuring balanced representation of responders and non-responders in each fold. For spatial transcriptomics datasets (e.g., breast, prostate), we adopt the patient-level splitting protocol proposed in \[2\]. Each split ensures that the test set contains patients not seen during training, thereby avoiding information leakage across spatial spots or slides derived from the same individual. This is critical to evaluating generalization performance. To evaluate generalization performance and mitigate overfitting, we use stratified splits across all tasks and report average metrics across multiple runs. Additionally, we employ regularization (e.g., dropout, weight decay) in all models. The consistent performance across held-out samples and repeated splits supports the claim that the models generalize well and are not overfitting to training data. In the PDO dataset, the sampling unit is cultures where each patient can contribute multiple organoid cultures, resulting in 1,678 cultures. Each culture is treated in triplicate with either vehicle control or titrated combinations of clinical therapies as described in \[3\]. We thank the reviewer for bringing this up and will update Appendix G to clarify these points.
>
> **Mis-aligned Appendix references**
>
> Thanks for pointing this out. We will fix this in the revised manuscript.
>
> We appreciate your feedback and hope that our responses have addressed all your concerns. Please let us know if there are any remaining questions or areas that would benefit from further clarification, we are happy to provide additional details. If you believe all your comments have been satisfactorily answered we would be grateful if you consider updating your score. We thank the reviewer again for your time and constructive review\!
>
> \[1\] Cai, C., & Wang, Y. (2020). A note on over-smoothing for graph neural networks. arXiv preprint arXiv:2006.13318
>
> \[2\] Wu, Z., Trevino, A. E., Wu, E., Swanson, K., Kim, H. J., D’Angio, H. B., ... & Zou, J. (2022). SPACE-GM: geometric deep learning of disease-associated microenvironments from multiplex spatial protein profiles. *bioRxiv*, 2022-05.
>
> \[3\] María Ramos Zapatero, Alexander Tong, James W. Opzoomer, Rhianna O’Sullivan, Ferran Cardoso Rodriguez, Jahangir Sufi, Petra Vlckova, Callum Nattress, Xiao Qin, Jeroen Claus, Daniel Hochhauser, Smita Krishnaswamy, and Christopher J. Tape. Trellis tree-based analysis reveals stromal regulation of patient-derived organoid drug responses. Cell, 186(25):5606–498 5619.e24, 2023\. ISSN 0092-8674.

---

> > ### Comment · Reviewer_kJJr · 2025-08-01
> >
> > I am satisfied with the response. Thank you for the additional oversmoothing benchmark and dataset clarifications

---

### Note · Authors · 2025-08-14

We thank reviewers for their valuable feedback. Some reviewer concerns included scalability, expressivity and the use of simplicial complexes over simple graphs, and clarification on single cell datasets, and cohorts and tasks.  We have addressed all these points thoroughly and summarize our responses below.

## Scalability and Computational Costs

To address computational costs associated with higher-order simplicial complexes and multiple views, we restrict the computation to up to second-order simplices and lower the $\epsilon$-thresholds to sparsify the complexes in our revised experiments, in which we retain strong performance while reducing our computational cost, allowing for application to large data sets. Our PyG implementation leverages batching and sparse tensors to reduce further overhead. Further we note below that these higher order constructs allow for more expressivity.

## Mitigating oversmoothing

To establish that we mitigate oversmoothing concerns prevalent in GNNs, we compute the Dirichlet Energy of the learned embeddings for all methods. HiPoNet achieved substantially higher energy than traditional message-passing networks, indicating mitigation of oversmoothing and preservation of high frequencies. This further validates our choice of incorporating multiscale wavelets into our architecture.

## Oriented vs Unoriented simplices

HiPoNet supports oriented and unoriented Laplacians, but in our experiments, we only use the unoriented Laplacian for all $k>0$. When $k=0$, we use oriented simplices, in which case the oriented Hodge Laplacian coincides with the graph Laplacian. The oriented case appears in Theorem 4.4 and Corollary 4.5, which address only the case of $k = 0$. Theorem 4.3, by contrast, holds without restriction to $k=0$ due to the structure of the simplicial graph (Definition 4.2), but its statement in the main text will be revised to point to Appendix D for a more detailed version.

## HiPonet is meant for high-dimensional point-cloud data cohorts

Reviewer WcXs asked about using basic ML methods like clustering. We clarified that our framework is designed for cases where observations consist of entire high dimensional point-cloud datasets of different sizes, and that we have many of these such that a batch forms a data cohort consisting of multiple data sets. Thus our framework is designed to make neural network classification and regression amenable for such high dimensional pointcloud datasets arising in biology.

---

### Decision · Program_Chairs · 2025-09-17

**Decision:**

Accept (poster)

**Comment:**

HiPoNet introduces a framework for analyzing high-dimensional point clouds using multi-view simplicial complexes and wavelet transforms, with a strong focus on biological datasets such as single-cell and spatial transcriptomics data.

The reviewers found the method well-motivated, empirically strong, and demonstrates significant performance gains over existing relevant baselines particularly in clinically relevant tasks. Reviewers appreciated the biological relevance, multi-view design, and ablation studies, while initial concerns around theoretical clarity and computational complexity were addressed sufficiently well in the rebuttal. Reviewers generally acknowledged the authors’ clarifications, and multiple scores were raised during discussion.

Overall, HiPoNet represents a technically solid and potentially impactful contribution to geometric deep learning and scientific machine learning. I therefore recommend acceptance with the expectation that the additional results and clarification that emerged during the discussion phase are included in the final version of the paper.